# Rapid ER remodeling induced by a peptide–lipid complex in dying tumor cells

Samudra Sabari, Siddharth Chinchankar, Ines Ambite, Atefeh Nazari, António Pedro NBM Carneiro, Axel Svenningsson, Catharina Svanborg, Arunima Chaudhuri

The membranous ER spans the entire cell, creating a network for the biosynthesis of proteins and lipids, cell-wide communication, and nuclear delivery of molecules, including therapeutic agents. Here, we identify a novel ER response triggered by the tumoricidal complex alpha1-oleate, defined by a loss of peripheral ER structure, extensive ER vesiculation. Alpha1-oleate was present in the ER-derived vesicle membranes, also decorated by ER-resident and ER-interacting proteins, calnexin and ORP3, and in their lumen, also enriched for KDEL, confirming their ER origin distinct from lipid droplets. Rapid nuclear uptake of the complex constituents resulted in diffuse nuclear staining, and the asymmetrical perinuclear enrichment of the collapsing ER with its content of alpha1-oleate created large invaginations lined by the ER, inner nuclear membrane markers, and lamin nucleoskeleton. In parallel, a change in nuclear shape resulted in a volcano-like structure. This newly discovered, potent ER response to alpha1-oleate may have evolved to package ER-associated cellular contents in the nuclei of dying tumor cells, thus sequestering toxic cell debris associated with apoptotic cell death.

## Introduction

The ER defines many aspects of cellular life. It is the largest membrane-bound organelle and accounts for more than 50% of the total membrane area in eukaryotic cells (Milo et al, 2010). Organized as a complex, continuous network of tubules and sheets, the ER spans the cytosol all the way from the cell periphery to the nuclei and plays a critical role as a scaffold for the organelles, with which it communicates, including ribosomes, mitochondria, and the Golgi (English et al, 2009; Pendin et al, 2011). The ER structure, spanning the cytosol, would be conducive to transporting molecules from the cell periphery into the nuclear ER, but this route of transport is not well documented, especially because entry of molecules into the ER is restricted. Transport of molecules from the cell periphery into the nuclei has mainly been studied for the endocytic pathway, and

escape from the endosomes is generally required for nuclear entry. Recently, perinuclear drug enrichment has been discussed as a mechanism for enhanced nuclear entry (Zhu et al, 2018; Zhang et al, 2019), suggesting a strategic potential of targeting the ER for nuclear delivery.

The highly membrane-interactive "HAMLET family" of complexes, formed by alpha-lactalbumin and oleic acid, shows potent tumoricidal activity against a broad range of tumor cells in vitro (Hakansson et al, 1995; Svensson et al, 2000; Svanborg et al, 2003; Ho et al, 2017), and a surprising degree of selectivity for tumor tissue in vivo in several cancer models (Fischer et al, 2004; Puthia et al, 2014; Tran et al, 2020). The alpha1-oleate complex, formed by the N-terminal alpha-helical peptide of alpha-lactalbumin, has been developed for clinical trials. Uptake of the alpha1-oleate complex by the tumor triggered rapid cell death by an apoptosis-like mechanism and a significant reduction in tumor size (Haq et al, 2024). Therapeutic efficacy was demonstrated in early HAMLET studies (Gustafsson et al, 2004; Mossberg et al, 2007) and for alpha1-oleate in a placebo-controlled trial of non–muscle-invasive bladder cancer (Tran et al, 2020; Brisuda et al, 2021), without drug-related side effects (Haq et al, 2024).

The sensitivity of tumor cells to the "HAMLET" family of complexes is guided by criteria generally accepted as "Hallmarks" of cancer (Storm et al, 2011; Ho et al, 2016). Oncogene signaling pathways are inhibited in alpha1-oleate–treated cells, and the expression of cancer-related genes is markedly reduced in treated cancer tissues (Tran et al, 2020; Brisuda et al, 2021). The uptake by tumor cells is followed by rapid transfer of the complex to the nuclear compartment, where interactions with histones and chromatin have been detected (Duringer et al, 2003). Uptake is insensitive to endocytic inhibitors, suggesting a different route of transport from the cellular periphery to the nuclei (Nadeem et al, 2019). Furthermore, the extent to which the complex remains intact once it enters tumor cells has not been clear, as technology to label both the peptide and lipid constituents of the complex has not been available. This study uses novel labeling technology to investigate whether the membrane-active alpha1-oleate complex targets the ER in tumor cells and whether this ER interaction might explain the efficient transport of

---

Division of Microbiology, Immunology and Glycobiology, Department of Laboratory Medicine, Lund University, Lund, Sweden

Correspondence: catharina.svanborg@med.lu.se

the complex from the cell periphery to the nuclei and the multitude of targets affected.

# Results

## Imaging of tumor cell entry and intracellular distribution of alpha1-oleate

The N-terminal domain of alpha-lactalbumin (alpha1) forms a complex with oleic acid (alpha1-oleate) that rapidly kills tumor cells. Cellular uptake of the complex has been inferred from imaging studies using the labeled peptide bound to oleic acid, but it has remained unclear to what extent both constituents of the complex enter tumor cells and interact with different tumor cell compartments. To address this question, the alpha1-peptide was N-terminally labeled with photostable dyes and oleic acid by click technology and a complex was formed from the labeled constituents (Fig S1A–E). The complex constituents JF549 alpha1-peptide and AF647-labeled oleic acid were traced intracellularly, by confocal imaging and Airyscan technology in A549 lung carcinoma cells (Figs 1, S2A–D, and S3A–C).

The alpha1-peptide and oleic acid were detected in the cytoplasm, perinuclear area, and nuclei of A549 cells after 10 min of exposure to alpha1-oleate, with a further increase after 20 and 60 min (Fig 1A–E, $P < 0.001$ for 20- and 60-min uptake compared with PBS). Co-localization of the alpha1-peptide and oleic acid was confirmed by line scans in the plasma membrane area, cytoplasm, perinuclear region, and nuclei (Figs 1B and D, S4A–F, and S5A–F), suggesting that both constituents of the complex reach these sites. Notably, a distinct punctate staining pattern of the alpha1 and oleate constituents was also observed in the cytoplasm and perinuclear regions. A similar staining pattern was detected in the U251 human glioblastoma cell line, the A498 human kidney cancer cell line (Fig 1H and I), and the HTB9 human urinary bladder cancer cell line (Fig S6A–C), confirming the uptake and subcellular distribution of alpha1-oleate in different cancer cells.

Uptake of the labeled complex was observed in 100% of the treated A549, U251, HTB9, and A498 cells (Fig S2A–D). In contrast, cells were not shown to internalize either the JF549-labeled alpha1-peptide or the AF647-labeled oleic acid, confirming that the ability to cross the plasma membrane and reach different cellular compartments is a characteristic of the alpha1-oleate complex (Figs 1F and G and S7A and B). The results suggest that the alpha1-oleate complex crosses the plasma membrane and reaches the interior of tumor cells, including the perinuclear and nuclear compartments.

## Rapid ER network remodeling and vesiculation

The perinuclear staining pattern, shown in Fig 1, suggested that the complex might accumulate in the ER. To address this hypothesis, the ER was labeled with a BODIPY-based ER-Tracker and the structure of the ER in A549 cells was imaged in real time, comparing alpha1-oleate–treated cells with the PBS, alpha1-peptide, or oleic acid as controls (Figs 2 and S8A–E). The peripheral ER was clearly visible in control cells, as a network of fine tubules and sheets with

visible three-way junctions (Fig 2A and B). In contrast, a rapid loss of peripheral ER structure was detected in alpha1-oleate–treated cells, resulting in an increasingly ER-depleted cellular periphery (Fig 2C and D). A loss of peripheral ER structure was detected in 100% of the cells after 10 min (Fig 2E; $P < 0.001$ compared with PBS) but not in cells treated with alpha1-peptide or oleic acid (Figs 2F and S8A–E).

The distance between the plasma membrane (PM) and the retracting ER network was quantified by drawing lines between the edge of the retracting ER network, defined by ER-Tracker staining, and the edge of the cell, defined by brightfield microscopy (63X magnification, Figs 2E and S9A–F). CellBrite staining was further introduced to visualize the PM, but CellBrite labeled a wider area in the cell periphery, which also included the peripheral ER. This wider staining at the cell periphery was lost in alpha1-oleate–treated cells, leaving a linear staining pattern at the border of the cell, interpreted as the PM, and a retracting and structurally chaotic pattern inside the cell, interpreted as the retracting ER (Figs 2G and H and S10A and B).

In parallel, the complex triggered the formation of ER-derived vesicle (EDV) membrane (Figs 3 and S11A). Small EDVs were observed in the cell periphery in 73% of the treated cells after 10 min with an increase to 89% after 60 min of alpha1-oleate exposure (Fig 3D; $P < 0.001$). With time, the EDVs increased in size, forming clusters surrounding the nuclei (Fig 3E and F). Co-localization of the labeled complex with the EDV membranes was detected, and the labeled complex was present inside the vesicles (Figs 3G and S11B and C). EDVs were not observed in cells treated with the alpha1-peptide or oleic acid (Fig S8C and D).

The loss of peripheral ER staining and an increase in EDVs were validated by live-cell imaging of A549 cells transfected with the ER marker halo-KDEL, which resides in the ER (Lukinavicius et al, 2013; Jiang et al, 2019) (Figs 3H–L and S12A and B). The halo-KDEL protein showed a similar change in distribution as the BODIPY-based ER-Tracker in cells treated with alpha1-oleate. To examine whether some of the vesicles were lipid droplets, KDEL-transfected cells were co-stained with silicon–rhodamine dye for ER visualization via halo-KDEL and LipidTOX Red Neutral Lipid Stain to label lipid droplets. Lipid droplets were detected in untreated cells, including at the cell periphery and the perinuclear region. Cells treated with alpha1-oleate showed a similar distribution of lipid droplets, with evidence of an increase in the number and size (Fig S13A–E). Interestingly, most of the EDVs stained by halo-KDEL were not stained by LipidTOX, demonstrating that the EDVs represent a separate class of vesicular structures (Figs 3M–O and S14A–C).

The molecular chaperone calnexin (Hebert & Molinari, 2007), which resides in the ER, showed a similar pattern as the ER-Tracker, with a loss of staining in the cell periphery and staining of the EDVs, as well as perinuclear accumulation (Fig S15A–D). Changes in calnexin distribution were detected in more than 90% of cells treated with alpha1-oleate after 60 min but not in cells treated with alpha1-peptide or oleic acid (Fig S15E–G).

The results identify a rapid change in ER structure triggered by alpha1-oleate and defined as peripheral ER collapse, ER membrane vesiculation, and perinuclear ER accumulation. A similar ER response was detected in the human glioblastoma cell line (U251), the human urinary bladder cancer cell line (HTB9), and the human

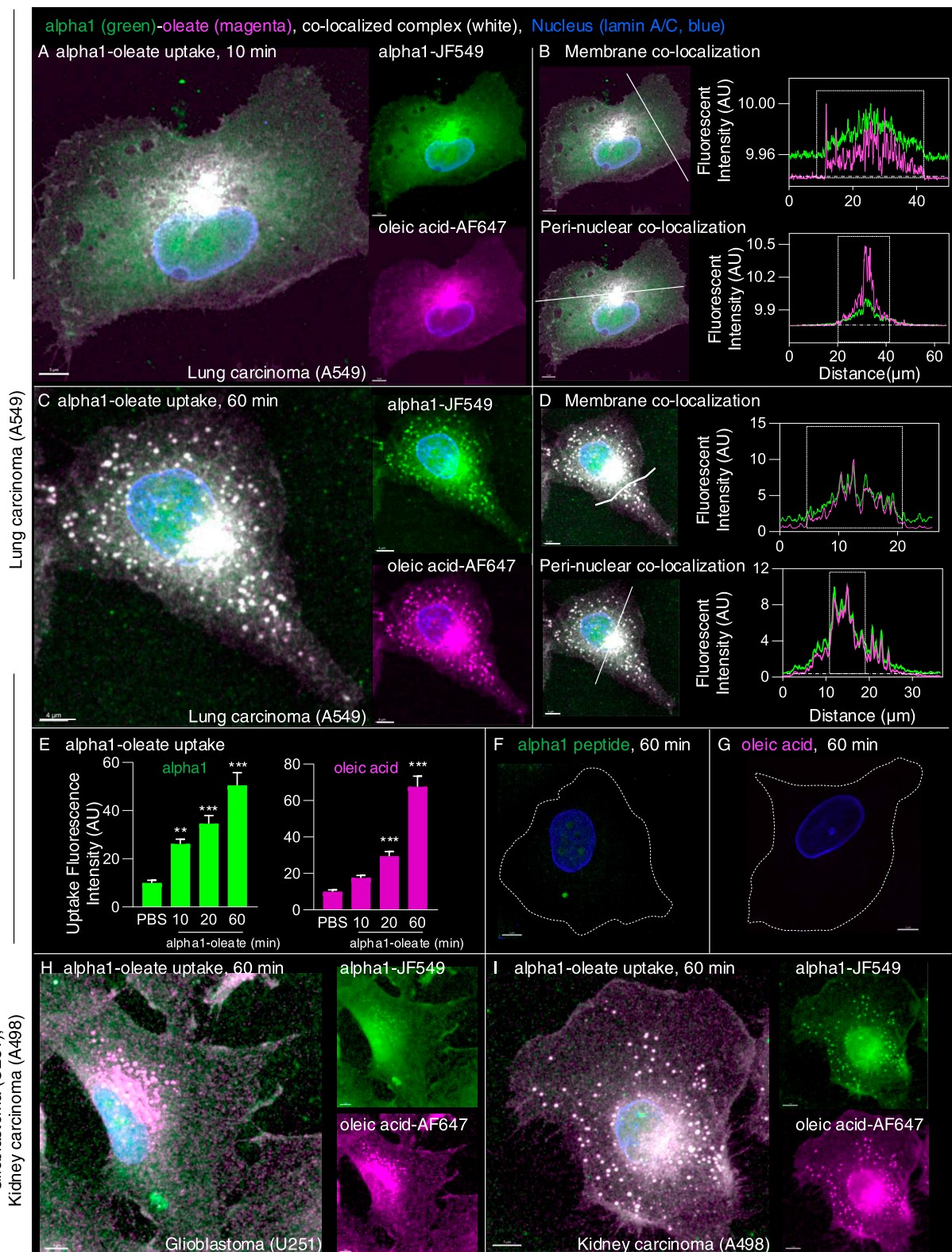

**Figure 1. Rapid internalization of the alpha1-oleate complex by tumor cells.**
**(A)** A549 cells were exposed to the alpha1-oleate complex formed by JF549-labeled alpha1-peptide (green) and the AF647 click–labeled oleic acid (magenta). Airyscan images of cells exposed to the labeled complex (35 $\mu M$) for 10 min, showing diffuse staining throughout the cytoplasm, intense staining in the perinuclear region, and staining inside the nuclei. Merged images and single channels are shown. In fixed cells, nuclei are visualized by lamin A/C (blue) immunostaining with anti-mouse AF405 secondary antibodies. **(B)** Line scans quantifying the alpha1-peptide and oleic acid signals in the cytoplasm, perinuclear, and nuclear areas after 10 min. **(C)** Distribution

kidney cancer cell line (A498) (Figs S16A–F and S17A–F). Notably, the ER-resident proteins calnexin and the KDEL marker were associated with the remodeled ER in alpha1-oleate–treated cells and EDVs formed in response to alpha1-oleate and shown to be distinct from lipid droplets.

### ER membrane interactions of alpha1-oleate

The ER membranes are relatively thin because of a unique lipid composition and loose lipid packing, and the membrane structure is highly dynamic, continuously undergoing significant remodeling to support the rapidly shifting demand for diverse molecular interactions (van Meer et al, 2008; Kucinska et al, 2023). Alpha1-oleate and HAMLET have been identified as highly membrane-active molecules, shown to trigger rapid blebbing and tubulation in model lipid membranes (Nadeem et al, 2015; Hansen et al, 2020). The observed effects of alpha1-oleate suggested that membrane interactions might be facilitated by the unique lipid composition of the ER membrane, where higher levels of phosphatidylcholine (PC) and phosphatidylethanolamine (PE) and lower levels of cholesterol and sphingolipids make the membrane thinner and more fluid than the PM (van Meer et al, 2008).

The effect of membrane composition on the interaction with alpha1-oleate was examined, using the giant unilamellar vesicle (GUV) model, where the response of lipid bilayers can be investigated, without the involvement of other resident membrane constituents. GUVs were formed using lipid mixtures representative of the ER membrane or the PM, and the response to alpha1-oleate was imaged in real time and quantified, compared with PBS controls. The ER membrane vesicles were significantly more responsive to alpha1-oleate, than the PM vesicles, defined by the frequency of GUV membrane vesiculation, tubulation, and division of vesicles, in response to alpha1-oleate treatment (Fig 4A–C). In contrast, control membranes exposed to PBS remained largely unchanged during the 60 min of observation.

The integration of alpha1-oleate into the GUV membranes was subsequently compared between GUVs composed of membrane ER lipids or PM lipids. The rhodamine-labeled GUVs were incubated with the AZ647-labeled alpha1-oleate. Co-localization of the complex with the rhodamine marker was observed in both types of GUVs, and a rapid membrane response with the formation of internal tubules and lysis was observed in the ER GUVs exposed to the labeled complex. In contrast, a delayed and a weaker response was seen in the PM GUVs (Fig S18A–D).

The results suggest that alpha1-oleate preferentially interacts with ER-like lipid bilayers and induces membrane vesiculation and tubulation, more efficiently than in PM-like bilayers. This ER vesiculation is consistent with observations in alpha1-oleate–treated cells.

### ER response to alpha1-oleate, defined by gene expression analysis and Western blots

The ER response to alpha1-oleate was further investigated by genome-wide transcriptomic analysis (Fig 4D–G). RNA was isolated from alpha1-oleate–treated A549 cells, and significantly regulated genes were identified, compared with untreated cells (absolute fold change [FC] > 1.5). Consistent with previous studies, the expression of cancer-related genes was inhibited in alpha1-oleate–treated cells and a broad apoptotic response was activated ($P$-value 1.2 × $10^{-47}$, 318 regulated genes), including *CASP9*, *BCL2A1*, *BCL3*, *BCL6*, *MCL1*, *MDM1*, *MDM4*, and NF-κB family genes (Fig 4G). Apoptosis was confirmed by TUNEL staining of alpha1-oleate–treated A549 cells (Fig 4H). Treatment with alpha1-peptide or oleic acid did not significantly affect gene expression (Fig S19A and B).

Functional analysis of regulated genes further identified the unfolded protein response and ER stress pathways as activated (Figs 4D and S19C; Z-score 2.89, $P$-value 6.9 × $10^{-10}$) (Figs 4E and S19D; Z-score 2.24, $P$-value 5.0 × $10^{-4}$), consistent with the observed change in ER structure and the uptake of the alpha1-oleate complex, in which the alpha1-peptide remains partially unfolded when bound to oleic acid (Brisuda et al, 2021). Significantly activated genes included heat shock protein (HSPs), *HSPA5*, *HSPA6*, and *HSPA7*, as well as *EIF2AK3*, *XBP1*, and *PPARG* (Fig S19C and D). Increased eIF2α phosphorylation after 1 h, detected by Western blot analysis, confirmed that the eIF2α arm of the ER stress response was activated (Fig 4F). The XBP1/IRE1 and ATF6 arms of the ER stress response were not significantly affected at the protein level (Figs 4F and S20A–E).

The response to alpha1-oleate was subsequently compared with known ER stress inducers tunicamycin and thapsigargin. The effects on ER structure in A549 were compared by live-cell imaging of cells stained with ER-Tracker (Figs S21A and B and S22A–C). In contrast to the rapid response in virtually all cells exposed to alpha1-oleate (60 min, Figs 2 and 3), there was no evidence of a loss of peripheral ER structure or EDV formation in most of the tunicamycin- or thapsigargin-treated cells after 12 h (Fig S21A and B). Destruction of ER ultrastructure was observed in 47% of the tunicamycin-treated cells after 12 h, and 23% of the thapsigargin-treated cells showed extensive, beehive-like deformations of the ER structure, resembling documented whorls (Xu et al, 2021). By Western blot analysis, a significant increase in p-eIF2α protein levels was detected in cells

of JF549-labeled alpha1-peptide and AF647 click–labeled oleic acid in A549 cells after 60 min of exposure. Airyscan images showing diffuse and punctate staining throughout the cytoplasm, intense staining in the perinuclear region, and staining inside the nuclei. Merged images and single channels are shown. **(D)** Line scans quantifying the alpha1-peptide and oleic acid signals in the cytoplasm, perinuclear, and nuclear areas after 60 min. Further line scans supporting the co-localization are shown in Figs S4 and S5. **(E)** Quantification of the complex constituents in whole cells. Data are expressed as the mean ± SEM from maximum intensity projections collected using z-stacks (n = 15 cells per time point) obtained from one typical experiment. Statistical significance was determined by one-way ANOVA with Šidák's multiple comparison test. ***$P$ < 0.001, **$P$ = 0.004. **(F, G)** Control experiment in A549 cells exposed to JF549-labeled alpha1-peptide or the AF647 click–labeled oleic acid. No evidence of uptake of the individual labeled complex constituents after 60 min. The white dotted line marks the cell periphery. **(H, I)** U251 glioblastoma cells and (I) A498 kidney carcinoma cells exposed to the alpha1-oleate complex formed by JF549-labeled alpha1-peptide and the AF647 click–labeled oleic acid. Airyscan images of cells exposed to the labeled complex (35 µM) for 10 min, showing diffuse and punctate staining throughout the cytoplasm, intense staining in the perinuclear region, and staining inside the nuclei. Merged images and single channels are shown. Scale bar, 5 µm (A, B, C, D, F, G, H); 7 µm (I).

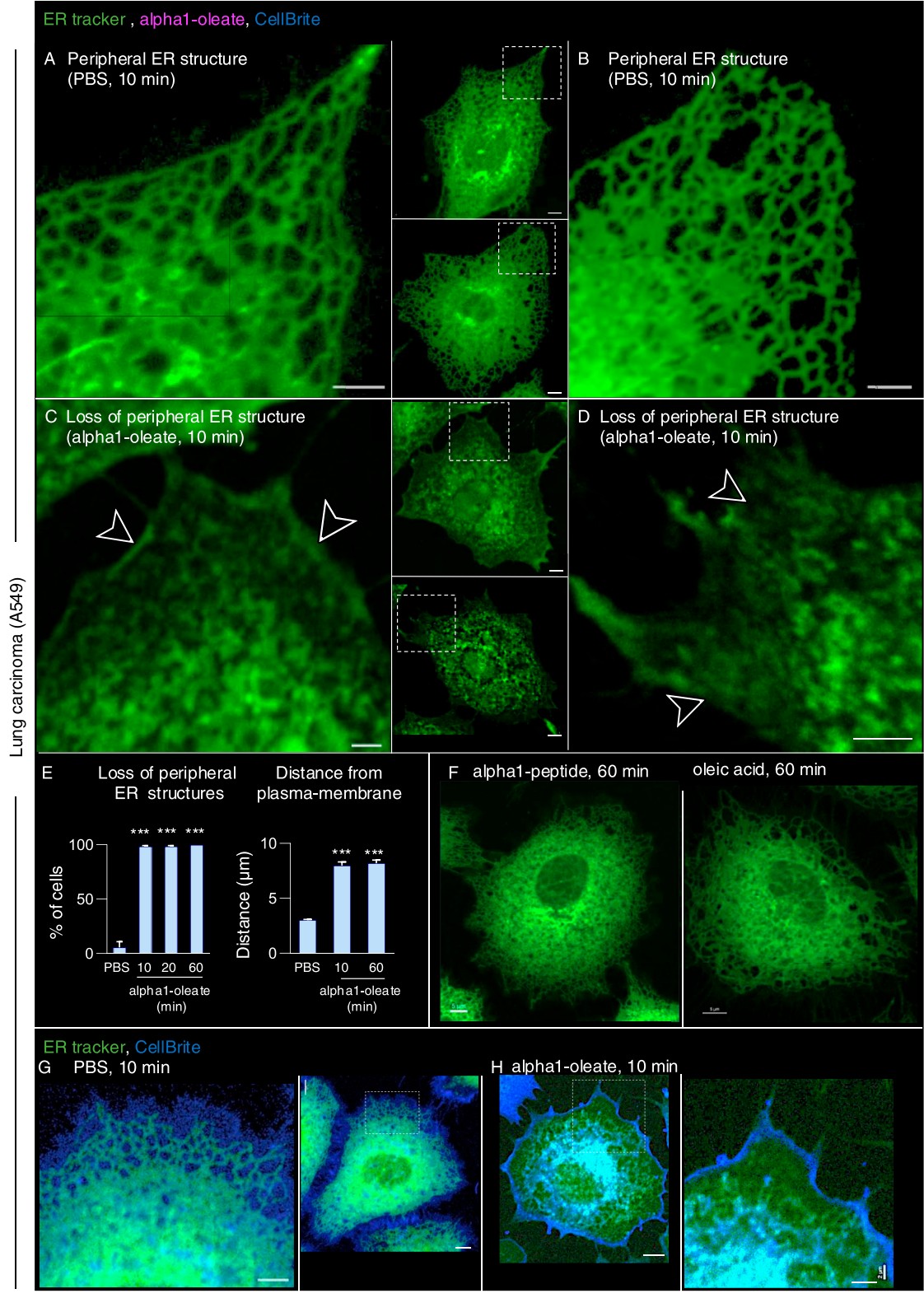

**Figure 2. Peripheral ER response induced by the alpha1-oleate complex.**
**(A, B)** Live-cell confocal images of A549 cells stained with the BODIPY-based ER-Tracker showing the ER network (green) extending from the cellular periphery to the perinuclear region in cells treated with PBS. **(C, D)** Rapid loss of peripheral ER structure in response to alpha1-oleate (unlabeled, 21 μM) treatment shown after 10 min of exposure. Loss of peripheral tubules and sheets in the retracting ER is indicated by arrows. **(E)** Quantification of the loss of peripheral ER structure and distance of the remaining ER from the cell periphery. Data are expressed as the mean ± SEM of three independent experiments, n = 50 cells per experiment. Statistical significance was

treated with tunicamycin for 12 h and an increase in p-eIF2α and IRE1α in thapsigargin-treated cells after 12 h, confirming the ER stress response (Figs S21E and F, S23A–E, and S24A–E). The activation of ER stress by tunicamycin and thapsigargin was confirmed by gene expression analysis (Figs S21 and S22). The ER stress response to alpha1-oleate was less extensive than that to tunicamycin or thapsigargin (Figs S21, S22, S23, and S24).

The results identify an active transcriptional response in alpha1-oleate–treated cells, defined by selective activation of programmed cell death, with an apoptosis-like profile and activation of the eIF2α arm of the ER stress response, in parallel with the inhibition of cancer-related gene expression in the tumor cells.

The comparative analysis suggested that the ER stress response triggered by alpha1-oleate is more rapid and restricted than the response elicited by thapsigargin and tunicamycin.

### ER-associated nuclear entry and NER formation triggered by alpha1-oleate

Nuclear uptake of the complex was detected in A549 lung carcinoma cells after 10 min, with an increase after 20 and 60 min of exposure (Fig 5A–C, $P < 0.001$ for 20- and 60-min uptake compared with PBS). Co-localization, demonstrated by line scans, supported the presence of both the complex constituents inside the nuclei (Figs S4A–F, S5A–F, and S25A–F). There was no evidence of significant cellular entry or nuclear translocation either of the labeled alpha1-peptide or of oleic acid (Fig S7A and B). Nuclear staining was largely diffuse and increased with time as shown by confocal imaging and z-stacks with intact or masked nuclei, using the lamin outline to define nuclear periphery (Figs 5A and B and S26A). The rapid nuclear entry of the labeled complex was not significantly affected by preloading WGA in A549 cells using the pinocytic method (Mohr et al, 2009; Konishi et al, 2017) to block nuclear entry (Fig S27A–E).

In addition, asymmetrical nuclear entry of the complex was observed from the perinuclear compartment, suggesting that the perinuclear ER creates a point of continued nuclear entry (Fig 5D and E). In Airyscan-based 3D reconstructions, large invaginations continuous with the perinuclear ER were shown to contain the peptide and oleic acid, suggesting that the complex may drive an expansion of the nucleoplasmic reticulum (NER) (Fig 5D and E). The reconstructions suggested that the perinuclear ER harboring the complex became continuous with the NER, apparently expanding the nuclear ER compartment. The invaginations projecting into the nuclei contained alpha1-oleate, as defined by the JF549-labeled alpha1-peptide and AF647-clicked oleate (Figs 5F and S26B–D).

The NER is formed by nuclear membrane invaginations into the nucleoplasm, and type II NER invaginations are lined both by the outer and by the inner nuclear membranes and may have cytosolic content, in contrast to type I invaginations, which only contain the inner nuclear membrane (Malhas et al, 2011; Drozdz & Vaux, 2017). The expansion of the nuclear ER was further probed by staining for the ER-resident protein calnexin and ER-interacting protein ORP3, a lipid transporter that interacts with the ER membrane (Weber-Boyvat et al, 2013). A significant time-dependent increase in nuclear calnexin and ORP3 content was detected in alpha1-oleate–treated cells (81% after 60 min, $P < 0.001$ compared with PBS; Figs 6A and B, S28A and B, and S29A–C). Calnexin and ORP3 staining marked the ridges of the invaginations, supporting the presence of the ER at this site (Figs 6A and B and S30A–E). Live-cell imaging of ER-Tracker–labeled cells confirmed an increase in nuclear ER content in alpha1-oleate–treated cells but not in cells exposed to the alpha1-peptide or oleic acid (Figs 6C–E and S31A–D), as well as tunicamycin or thapsigargin (Figs S21A and B and S22A–C). No change in ORP3 cellular distribution was observed in cells treated with the alpha1-peptide or oleic acid as controls (Fig S32A–D). To investigate whether the nuclear entry of the ER was associated with the alpha1-oleate complex constituents, A549 cells were treated with labeled complex and co-stained with calnexin and lamin A/C. Airyscan imaging coupled with 3D reconstructions revealed the nuclear invaginations continuous with the perinuclear ER, marked with calnexin, and associated with the complex constituents (Figs 5F–I and S33A and B).

These results suggest that there are at least two mechanisms leading to nuclear entry of the complex. One is detected as diffuse staining of the peptide and oleic acid, symmetrical and apparently unrelated to NER formation. The second phase of entry is marked by an increase in NER formation. The complex enters the nucleus encapsulated within the NER, but it remains unclear whether it ultimately reaches the nuclear lumen or remains enclosed within the ER as nuclear changes progress. Nonetheless, the NERs and the complex inside the NERs are clearly shown to be located inside the nuclear invaginations.

### Change in nuclear shape

The nuclear accumulation of alpha1-oleate and expansion of the NERs were accompanied by a significant change in nuclear shape (Fig 7A–E). Solid 3D renderings of the nucleus with lamin immunostaining clearly showed a change from a smooth, ellipsoidal shape to a volcano-like nuclear shape with large invaginations predicting points of nuclear ER entry (Figs 7A and B and S34A–D). A significant increase in nuclear height was observed, and concavity analysis confirmed the change in surface

---

determined by one-way ANOVA with Šidák's multiple comparison test for loss of peripheral ER structure and the Kruskal–Wallis test with Dunn's multiple comparison test for distance of ER from periphery. ***$P < 0.001$. **(F)** Control experiments in A549 cells exposed to alpha1-peptide (21 $\mu$M) or oleic acid (105 $\mu$M) show no change in peripheral ER structure. **(G, H)** Membrane marker CellBrite (blue) was used to visualize the change in membrane staining at the cell periphery of alpha1-oleate–treated cells. Scale bar, 3 $\mu$m ((A, B, C, D, G, H): cell periphery), 5 $\mu$m ((A, B, C, D, F, G): whole cell, (H): whole cell).

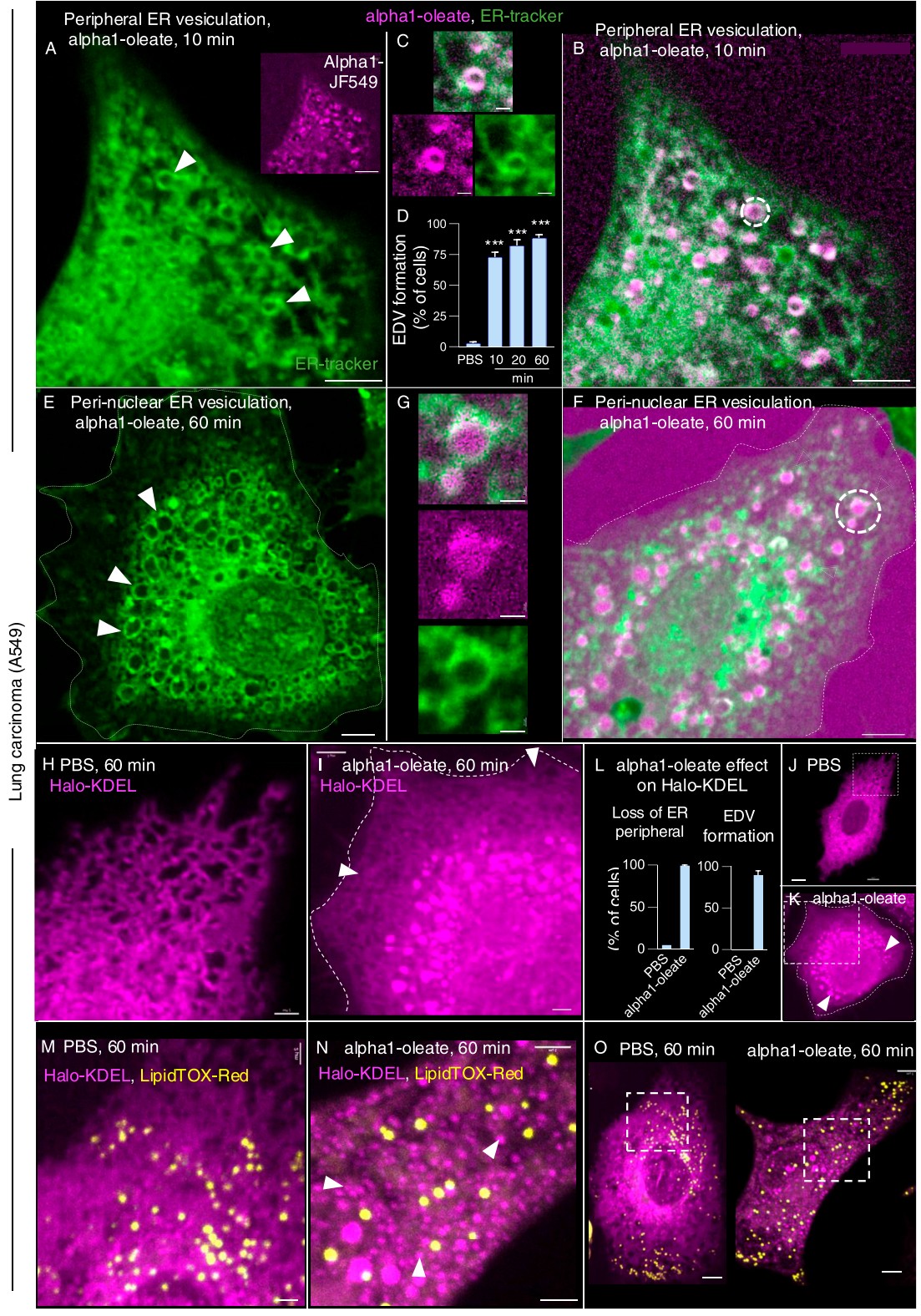

**Figure 3. Formation of ER-derived vesicles (EDVs) in alpha1-oleate–treated cells.**
**(A, B)** Live-cell confocal images of A549 cells exposed to alpha1-oleate for 10 min. The AZ647-labeled peptide is shown in magenta. EDV formation was observed in the cell periphery after 10 min of exposure to the labeled alpha1-oleate (mixed complex). ER membranes were stained with ER-Tracker (green), and examples of EDVs are indicated by arrows. **(B, C)** Zoomed-in image of an EDV (in (B)) showing the co-localization of alpha1-oleate with the EDV membrane and presence inside the lumen of vesicle. **(D)** Quantification of EDV formation triggered by alpha1-oleate (unlabeled, 21 $\mu$M). Data are expressed as the mean ± SEM of three independent experiments,

curvature over time as an increase in the ratio of concave to total surface area (Fig 7D; $P < 0.001$ for 40 and 60 min). Concavity analysis was performed using nuclear z-stacks, reconstructed in MATLAB (Biedzinski et al, 2020). No change in nuclear structure was detected in cells treated with alpha1-peptide or oleic acid as controls (Fig S35A–D).

The response to alpha1-oleate was further investigated by staining for SUN1 and SUN2, which are integral transmembrane proteins present in the inner nuclear membrane and interacting with the lamin nucleoskeleton, on the nucleoplasmic face of the nuclear membrane (Haque et al, 2010). Solid 3D renderings of the nucleus with SUN1 and SUN2 fluorescence signals confirmed the change in shape of the inner nuclear membrane (Figs 7A–C, S36A–D, and S37A–D), characterized by an increase in height, the formation of ridges, and major invaginations (Figs 7E, S38A and B, and S39A–D; $P < 0.001$ compared with PBS) in contrast to the more flat, ellipsoidal shape of the nuclei in untreated cells.

These results identify a change in nuclear shape involving organized multi-membrane assemblies containing alpha1-oleate and formed by the ER and proteins resident in the inner nuclear membrane and the nucleoskeleton.

### Disruption of the microtubular network

The change in nuclear shape suggested that the interaction with the cytoskeleton, which maintains the extended shape of the nuclei, might be affected by alpha1-oleate. This question was addressed by staining for α-tubulin, which is a major microtubular constituent (Fig 7F–H). A significant decrease in cellular α-tubulin staining intensity was observed in alpha1-oleate–treated cells (46.2% compared with PBS, $P < 0.001$) (Figs 7H and S40A and B). The perinuclear α-tubulin staining was reduced in alpha1-oleate–treated cells, and the dense microtubular network enveloping the nucleus was replaced by a tangled network with a loss of nuclear connectivity (Figs 7G and H and S40A and B). No significant effect on the microtubular network was observed in cells treated with the alpha1-peptide or oleic acid as controls (Fig S40C–E). Significant effects of alpha1-oleate on nesprin structure were not detected (Fig S41A–E).

The results suggest that disruption of the microtubular network by alpha1-oleate treatment contributes to the change in nuclear shape.

## Discussion

This study identifies a novel ER response, defined by peripheral ER collapse, extensive vesiculation, and expansion of the nuclear ER from the perinuclear area. The integration of alpha1-oleate into the ER membrane is proposed to gradually change its characteristic peripheral structure, from a network of fine tubules and sheets with visible three-way junctions, to a more chaotic structure that retracts toward the cell interior, accumulates in the perinuclear area, and invades the nuclear ER that it expands. The formation of ER-derived vesicles further supported a change from an extended, more linear membrane structure to one of the vesicles, many of which contained the alpha1-oleate complex in the membrane and the vesicle lumen. Such pronounced ER re-organization, involving the entire cell population, has not previously been described. The expansion of the nuclear ER also created a joint space inside the nuclei, containing the complex and ER-derived proteins, and was lined from the nuclear side by nuclear membrane proteins and the nucleoskeleton. This new compartment may serve to sequester cargo of dying tumor cells until the cells are removed and "cleared/expelled" from the tumor tissue.

Peripheral ER collapse has not been described in human cells, but in yeast, a fraction of the cortical ER detaches from the PM during cell division and forms a spatially distinct compartment adjacent to the nuclei (Otto et al, 2021). Here, the rapid peripheral ER collapse was further accompanied by the formation of ER membrane vesicles of gradually increasing sizes, surrounding the nucleus. ER collapse and vesiculation are not a characteristic feature of the ER stress response, as agents known to induce an ER stress response such as tunicamycin or dithiothreitol have not been shown to induce ER vesicle formation (Lu et al, 2013). This finding was further validated by our results, as extensive ER swelling and dilation were induced by tunicamycin treatment, ultimately leading to ER fragmentation, consistent with previous observations (Oslowski & Urano, 2011). In contrast, the formation of convoluted, ring-like structures known as ER whorls was observed after thapsigargin treatment, which disrupts calcium homeostasis by inhibiting the sarcoplasmic/ER $Ca^{2+}$-ATPase (Xu et al, 2021). ER remodeling into micrometer-scale intracellular vesicles has been described in cells exposed to hypotonic buffer defined by the retention of the luminal KDEL protein (King et al, 2020). ER vesiculation was also described in cells exposed to the small molecule dispergo (Lu et al, 2013). The ER effects of the

---

n = 50 cells per experiment. Statistical significance was determined by one-way ANOVA with Šidák's multiple comparison test. ***$P < 0.001$. **(E, F)** Live-cell confocal images of A549 cells exposed to unlabeled alpha1-oleate (21 µM) or labeled alpha1-oleate (mixed complex) for 60 min. Clusters of EDVs in the perinuclear area are indicated by white arrows. **(F, G)** Zoomed-in image of an EDV (in (F)) showing the co-localization of alpha1-oleate with the EDV membrane (merged image), the peptide (magenta), and ER-Tracker (green). The peptide was detected in the ER membrane and inside the lumen of vesicle. **(H, I)** Halo-KDEL–transfected A549 cells, confirming the loss of peripheral ER structure (arrows) after 60-min exposure to alpha1-oleate (unlabeled, 21 µM) compared with PBS. Cells were counterstained with silicon–rhodamine dye (magenta). Representative live-cell confocal images. **(J, K)** Halo-KDEL–transfected A549 cells show the formation of EDVs after 60-min exposure to alpha1-oleate (unlabeled, 21 µM) compared with PBS treatment. **(L)** Quantification of loss of periphery and EDV formation triggered by alpha1-oleate (unlabeled, 21 µM) is provided. Data are expressed as the mean of two independent experiments, n = 40 cells per experiment. **(M)** Distribution of lipid droplets in A549 cells transfected with Halo-KDEL. The ER network was visualized with silicon–rhodamine dye (magenta), and the lipid droplets by counterstaining with LipidTOX Red (yellow). **(N)** Most of the halo-KDEL–filled EDVs (magenta, arrows) formed upon exposure to alpha1-oleate (unlabeled, 21 µM) for 60 min are not co-localized with lipid droplets. Live-cell confocal image, representative cells are shown. **(M, N, O)** Confocal images of the whole cell magnified in (M, N) are provided. Mixed alpha1-oleate: 1:1 vol/vol, labeled, 35 µM, and unlabeled, 21 µM. Scale bar, 5 µm (A, B, E, F, J, K, O); 2 µm (H, I, M, N); 1 µm (C, G).

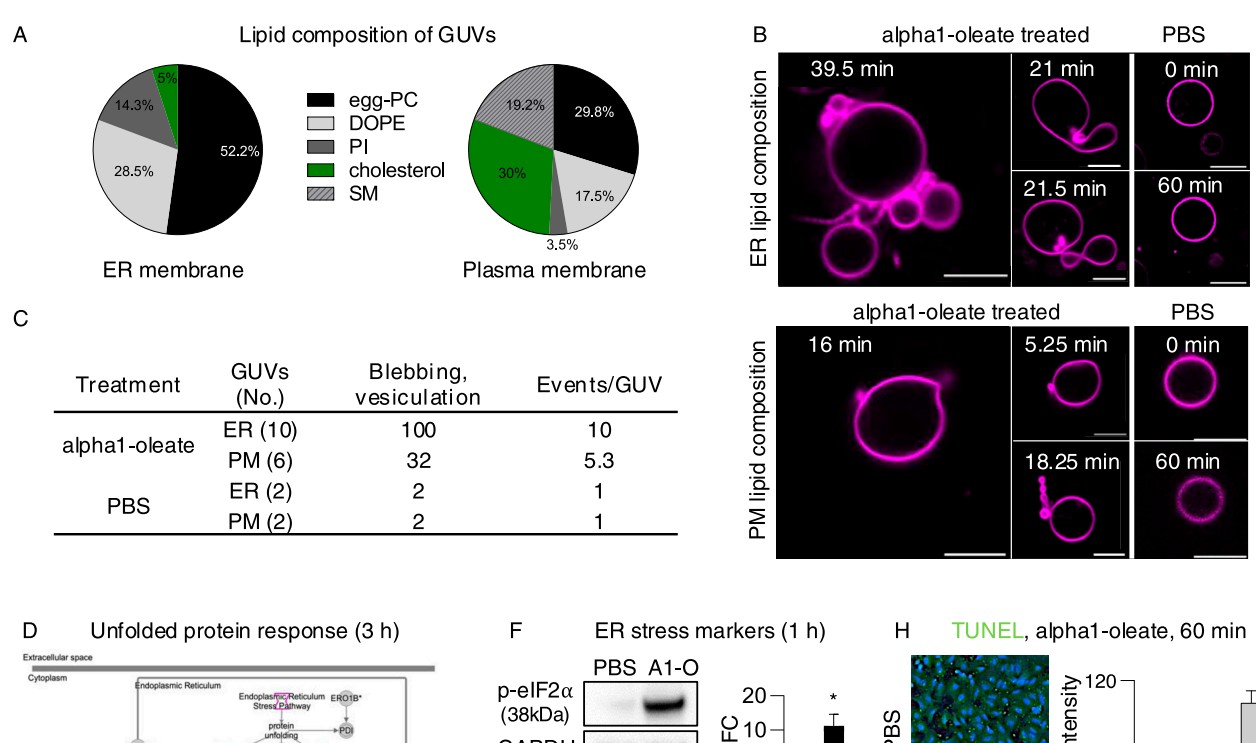

**Figure 4. ER response defined by membrane composition and defined by gene expression analysis.**
**(A)** Membrane vesicles (GUVs) were prepared using lipid mixtures representative of the ER or plasma membranes (PM) and exposed to alpha1-oleate (unlabeled, 21 μM) or PBS. GUVs were visualized using rhodamine B (magenta). **(B)** Rapid formation of membrane vesicles in ER-like GUVs exposed to alpha1-oleate (unlabeled, 21 μM). Minor structural changes were observed in PM-like GUVs exposed to alpha1-oleate. Stable morphology in GUVs exposed to PBS for both ER- and PM-like composition.

membrane-active alpha1-oleate complex were more extensive, however, involving the entire cell population and driving a pronounced ER reorganization in tumor cells, which has not previously been described.

This study detected a preferential interaction of alpha1-oleate with the ER membrane, compared with the PM in the GUV model. The preferential interaction with ER was attributed to differences in membrane lipid composition. The ER membrane has the lowest percentage of cholesterol (van Meer et al, 2008), a property that facilitates the rapid membrane insertion of alpha1-oleate followed by vesiculation and tubulation (Hansen et al, 2020). In previous studies, membrane insertion of alpha1-oleate and blebbing of the PM have been detected in response to alpha1-oleate (Nadeem et al, 2015) in tumor cells and GUV membranes composed of single phospholipid species such as phosphatidylcholine, suggesting that a lipid bilayer is sufficient for the membrane response to occur (Hansen et al, 2020). With prolonged exposure, the GUVs may be converted into bundles of tubuli or collapse to form membrane debris. The complex triggers rapid ion fluxes across cellular membranes (Storm et al, 2011; Nadeem et al, 2019). The exact mechanism of membrane integration of the complex is not yet known, but the pattern is not suggestive of leakage or pore formation in the GUV model (Nadeem et al, 2019). Previous studies have investigated the effect of membrane lipid composition on the response to alpha1-oleate. For example, the addition of cholesterol has been shown to reduce the membrane response to the complex (Ho et al, 2022).

In addition to the effects on the ER membrane, we detected potent effects of alpha1-oleate on the nuclear membrane, also involving the inner nuclear membrane constituents SUN1 and SUN2 and lamin nucleoskeleton. Oleic acid has previously been shown to induce intranuclear tubulation in Chinese hamster ovary K1 cells through a mechanism that requires the activation and reversible translocation of lipid synthesis enzyme CTP:phosphocholine cytidylyltransferase-$\alpha$ (CCT-$\alpha$), whose membrane-binding domain was shown to be necessary for NER expansion (Lagace & Ridgway, 2005; Gehrig et al, 2008). In addition, siRNA knockdown experiments revealed that *LMNA* expression was required for oleate-stimulated NER proliferation, consistent with our findings of lamin participating in the nuclear shape change. The alpha1-oleate complex was shown to deliver oleic acid to the nuclear membrane and nuclear interior, where it was shown to co-localize with its peptide partner, alpha1. The more extensive effects on nuclear membrane structure observed here, and the extensive ER vesiculation, suggest that both complex constituents are critically important for the nuclear membrane effects of the complex and that neither alpha1 nor oleic acid alone reproduces these effects in treated tumor cells. Higher concentrations of oleic acid, similar to those used in previous studies, did not recreate the extensive response to alpha1-oleate that occurred in the entire cell population.

Our results suggest that the alpha1-oleate complex, which belongs to the HAMLET family, retains itself inside the dying cancer cell nuclei. This compartmentalization might serve to reduce the leakage of cancer-promoting cell debris. Standard cytotoxic cancer therapy creates by-products consisting of killed tumor cell debris that stimulate primary tumor growth by triggering proinflammatory macrophage responses (Jiang et al, 2020). In addition, chromatin complexes expelled from nuclei of apoptotic cancer cells activate the RAGE receptors on neighboring surviving cancer cells, boosting the expression of metastasis-associated protein S100A4, which enhances their migration and invasiveness (Park et al, 2023). This may be due in part to the high affinity of alpha1-oleate for histone H3 and chromatin and spliceosome constituents, which has earlier been shown for the HAMLET complex (Duringer et al, 2003). Once it enters the nucleus, it may further trap the nuclear contents including the chromatin in the packaging compartment and prevent their escape from the cell's body and consequently the noxious after-effects of cytotoxic cancer therapies.

Alpha-lactalbumin is an ancient protein that defines lactation and the survival of mammals. Its close relative lysozyme is essential for the antimicrobial defense of early life-forms, and alpha-lactalbumin arose from lysozyme after a series of gene duplications 300–400 million yr ago, placing this family of proteins in an early evolutionary context (Hansen et al, 2020). The native protein is essential for lactose synthesis, which is required for the expression of milk from the mammary gland (Svanborg et al, 2003). The structurally flexible peptide forms complexes with oleic acid, and this study suggests that their affinity for lipid membranes allows rapid targeting of ER and perinuclear enrichment. We identify a unique ER-driven cell death program triggered by components found in mother's milk such as the HAMLET family of complexes that may have evolved naturally to package the cellular contents in the nucleus of rounded cells, to facilitate targeting and removal of immature cells, virus-infected cells, and emergent tumor cells from infant tissues exposed to the contents of human milk. Clinical and animal model studies of bladder cancer demonstrate the tumor specificity and a lack of side effects of alpha1-oleate, highlighting this proposed evolutionary selectivity for immature cells and cancer progenitors in the digestive tract of babies.

---

**(B, C)** Quantification of membrane effects in (B) compared with PBS is provided. **(D, E)** Whole-genome transcriptomic analysis of human lung epithelial cells treated with alpha1-oleate (35 $\mu$M, 3 h) compared with PBS (control) (cutoff fold change ≥ 1.5). **(D, E)** Activation of the unfolded protein response pathway (D) and ER stress pathway (E) in response to alpha1-oleate treatment shown (red = up-regulation, blue = down-regulation, orange = predicted activation, light blue = predicted inhibition, gray = not significantly regulated). **(F)** Western blot analysis of top regulated ER stress markers, phosphorylated eIF2$\alpha$ and XBP1S in alpha1-oleate–treated cells (1 h) compared with PBS-treated controls. GAPDH is shown for loading control. Data are expressed as the mean of three independent experiments. Statistical significance was determined by a two-tailed unpaired $t$ test. *$P$ < 0.03, ns, not significant. **(G)** Strong up-regulation of apoptosis-related genes in alpha1-oleate–treated cells (318 genes compared with untreated controls). Top regulated apoptosis signaling genes are shown in the table. **(H)** TUNEL staining demonstrates apoptosis-like cell death triggered by alpha1-oleate, n = 50 cells (H). Scale bar, 10 $\mu$m (B); 30 $\mu$m (H).

# Materials and Methods

**Key resources table**

| Reagents and resources | Source | Catalog number |
|---|---|---|
| Antibodies | | |
| Rabbit anti-lamin A/C IF: 1:200 | Abcam | ab224816 |
| Mouse anti-lamin A/C IF: 1:200 | Santa Cruz | sc-376248 |
| Mouse anti-Orp3 IF: 1:50 | Santa Cruz | sc-398326 |
| Mouse anti-giantin IF: 1:500 | Abcam | ab37266 |
| Rabbit anti-calnexin IF: 1:400 | Abcam | ab22595 |
| Rabbit anti-TGN46 IF: 1:100 | Novus Biologicals | NBP1-49643SS |
| Rabbit anti-SUN1 IF: 1:50 | Sigma-Aldrich | HPA008461 |
| Rabbit anti-SUN2 IF: 1:50 | Sigma-Aldrich | HPA001209 |
| Rabbit anti-atlastin1 IF: 1:50 | MyBioSource | MBS8245407 |
| Rabbit anti-nesprin 2 IF: 1:200 | Novus Biologicals | NBP2-38620 |
| Rabbit anti-reticulon 1 IF: 1:50 | MyBioSource | MBS2033943 |
| Mouse anti-alpha1-tubulin IF: 1:300 | Novus | NB100-690SS |
| Rabbit anti-RPL3 IF: 1:100 | Proteintech | 11005-1-AP |
| Rabbit anti-IRE1alpha (14C10) WB: 1:1,000 | Cell Signaling | 3294S |
| Rabbit anti-p-eIF2alpha (S51) WB: 1:1,000 | Cell Signaling | 3597S |
| Rabbit anti-XBP1S WB: 1:1,000 | Proteintech | 24868-1-AP |
| Mouse anti-ATF6 WB: 1:500 | Abnova | MAB6762 |
| GAPDH HRP WB: 1:4,000 | Santa Cruz | sc.25778 |
| Goat anti-mouse DyLight 405 IF: 1:200 | Invitrogen | 35501BID |
| Goat anti-rabbit DyLight 405 IF: 1:200 | Invitrogen | 35551 |
| Goat anti-mouse Alexa Fluor 488 IF: 1:200 | Invitrogen | A32723 |
| Goat anti-rabbit Alexa Fluor 488 IF: 1:200 | Invitrogen | A11034 |
| Goat anti-mouse Alexa Fluor 647 IF: 1:200 | Invitrogen | A32728 |
| Goat anti-rabbit Alexa Fluor 647 IF: 1:200 | Invitrogen | A32733 |
| Goat anti-rabbit IgG (H + L)-HRP conjugate WB: 1:4,000 | Bio-Rad | 1706515 |
| Rabbit anti-mouse immunoglobulins/HRP WB: 1:4,000 | Agilent | P0260 |
| Chemicals, peptides, and recombinant proteins | | |
| Sodium oleate | Sigma-Aldrich | O7501 |
| AF647-azide | Life Technologies | A10277 |
| AF488-azide | Jena Biosciences | CLK-1275-1 |
| ATPlite | PerkinElmer | 6016947 |
| PrestoBlue Cell Viability Assay | Invitrogen | A13262 |
| μ-Slide VI 0.4 | ibidi | 80606 |
| μ-Slide I | ibidi | 80106 |
| poly-L-lysine hydrobromide | Sigma-Aldrich | P2636-25MG |
| Goat serum | Dako | X090710-8 |
| ProLong Glass Antifade Mountant | Invitrogen | P36980 |
| WGA 488 | Invitrogen | W6748 |
| Click-iT cell Reaction Buffer Kit | Thermo Fisher Scientific | C10269 |
| Click-iT TUNEL Alexa Fluor 488 imaging assay kit | Thermo Fisher Scientific | C10245 |

**Continued**

| Reagents and resources | Source | Catalog number |
|---|---|---|
| Hoechst 33342 | Molecular Probes | H1399 |
| Janelia Fluor (JF) 549 NHS ester | TOCRIS | 6147 |
| AZDye 647 NHS Ester | Click Chemistry Tools | 1,344-1 |
| Coverslip Sealant | Biotium | 23005 |
| D-Tube Dialyzer Midi | Merck Millipore | 71506 |
| Amicon Ultra-15 Centrifugal Filter Unit | Merck Millipore | UFC900324 |
| ER-Tracker Green (BODIPY FL Glibenclamide) | Thermo Fisher Scientific | E34251 |
| Halo-KDEL plasmid was a gift from Jin Wang (http://n2t.net/addgene:124316; RRID:Addgene_124316) | Addgene | 124316 |
| Amaxa Nucleofection kit | Lonza | VCA-1002 |
| CellBrite Fix 555 Membrane dye | Biotium | 30088A |
| Sodium pyruvate | Gibco | 11360070 |
| Gentamicin | Gibco | 15750060 |
| MEM NEAA | Gibco | 11140050 |
| FBS | Cytiva | SV30160.03 |
| RPMI 1640 medium | Cytiva | SH30027.01 |
| alpha1-peptide (Ac-KQFTKAELSQLLKDIDGYGGIALPELIATMFHTSGYDTQ-OH) | Mimotopes | Australia |
| Chloroform | Merck | 102444 |
| Cholesterol | Sigma-Aldrich | C4951 |
| Rhodamine B (1 mg/ml) | Sigma-Aldrich | 83689 |
| Type IX-A agarose | Sigma-Aldrich | A2576 |
| AttoFluor cell chambers | Thermo Fisher Scientific | A7816 |
| 96-well plates | Corning | 3610 |
| Trypan blue solution | Sigma-Aldrich | T8154 |
| Paraformaldehyde, 16% solution in water | Electron Microscopy Sciences | 15710 |
| Glutaric dialdehyde, 25% solution in water | Electron Microscopy Sciences | 16220 |
| Nonidet P-40 Substitute | Sigma-Aldrich | 74385 |
| Triton X-100 | Alfa Aesar | A10646 |
| 20X TBS buffer | Thermo Fisher Scientific | 28358 |
| Tween-20 | Sigma-Aldrich | P1379 |
| Live-cell imaging solution | Invitrogen | A14291DJ |
| Pierce RIPA Buffer | Thermo Fisher Scientific | 89901 |
| Pierce 660 nm Protein Assay Reagent | Thermo Fisher Scientific | 22660 |
| Loading buffer | Invitrogen | NP0007 |
| SDS–PAGE | Invitrogen | BN1002B0X |
| Trans-Blot Turbo Transfer Pack | BIO-RAD | 1704156 |
| PhosSTOP EASYpack | Roche | PHOSS-RO |
| cOmplete Tablets EASYpack | Roche | 04693116001 |
| Novex ECL substrate | Invitrogen | WP20005 |
| Sucrose | Sigma-Aldrich | S0389 |
| Glucose | Thermo Fisher Scientific | A16828.36 |
| Fluoromount aqueous mounting media | Sigma-Aldrich | F4680 |

**Continued**

| Reagents and resources | Source | Catalog number |
|---|---|---|
| RNeasy Mini Kit | QIAGEN | 74104 |
| QIAshredder | QIAGEN | 79654 |
| GeneChip 3'IVT PLUS Kit | Thermo Fisher Scientific | 902416 |
| RNAlater using the AllPrep DNA/RNA/miRNA Universal Kit | QIAGEN | 80224 |
| Qproteome Cell Compartment Kit | QIAGEN | 37502 |
| Influx Pinocytic Cell-Loading Reagent | Molecular Probes | I-14402 |
| LipidTOX Red Neutral Lipid Stain | Thermo Fisher Scientific | H34476 |
| Tunicamycin | MedChemExpress | HY-A0098 |
| Thapsigargin | MedChemExpress | HY-13433 |
| (Z)-octadec-9-en-17-ynoic acid | Avanti Polar Lipids | 900412 |
| Egg phosphatidylcholine (Egg-PC) | Avanti Polar Lipids | 840051C |
| 18:1 PS (DOPS) | Avanti Polar Lipids | 840035C |
| Dioleoylphosphatidylethanolamine (DOPE) | Avanti Polar Lipids | 850725C |
| Phosphatidylinositol (PI) | Avanti Polar Lipids | 850142P |
| 15:0 PC | Avanti Polar Lipids | 850350C |
| 22:6 PC | Avanti Polar Lipids | 850400C |
| 24:1 sphingomyelin | Avanti Polar Lipids | 860593P |

## Methods and protocols

### *N-terminal labeling of alpha1-peptide with fluorophores*

The N-terminal labeling of the peptide with JF549 and AZ647 with the NHS functional group was performed according to the manufacturer's instructions. The free dye was removed using dialysis cassette (D-Tube Dialyzer Midi) followed by centrifugation with Amicon centrifugal filter unit (Amicon Ultra-15). The concentration of the labeled complex was determined using a Cary UV-Visible spectrophotometer or NanoDrop 2000c (Thermo Fisher Scientific).

### Peptide synthesis and complex generation

Peptides for in vitro and clinical experiments were synthesized using Fmoc solid-phase chemistry (Mimotopes). A fivefold stoichiometric concentration of sodium oleate in phosphate-buffered saline was prepared, followed by the addition of the peptide for preparation of the unlabeled complex. The preparation of the labeled complex is described in Appendix Fig S1.

### Cell lines and cell culture

Human lung carcinoma cells A549 (Cat# CCL-185), human glioblastoma (U251, gift from Prof. Alexander Pietras (Cat# 09063001)), urinary bladder cancer HTB-9 (Cat# 5637), and human kidney cancer A498 (Cat# HTB-44) were obtained from the ATCC. The cells were cultured in RPMI 1640 or DMEM with nonessential amino acids (1:100), 1 mM sodium pyruvate, 50 $\mu$g/ml gentamicin, and 5% or 10% FBS at 37°C, 5% $CO_2$.

## Cell viability assays

To quantify effects on cell viability, A549 cells were seeded in 96-well plates (2 × 10$^4$/well), cultured overnight at 37°C, 5% CO2, and incubated with the alpha1-oleate complex in serum-free RPMI 1640 at 37°C. FBS was added after 1 h, and cell viability was quantified after 5 min for the labeled complex by measuring cellular ATP levels using luminescence-based ATPlite kit and microplate reader (Infinite F200; Tecan).

Cell death was quantified by two biochemical methods: cell viability was quantified via PrestoBlue fluorescence and cellular ATP levels via a luminescence-based ATPlite kit. Fluorescence and luminescence were measured using a microplate reader (Infinite F200; Tecan).

Cell viability was also monitored with trypan blue exclusion assay. A549 cells (in suspension) were added to a 96-well plate (5 × 10$^4$/well). Then, cells were treated with alpha1-oleate for 1 h. Trypan blue (0.4%) was added in a 1:1 ratio with the single-cell suspensions, and the cells were monitored and quantitated using the Bürker chamber under a brightfield microscope.

## Uptake of alpha1-oleate and immunofluorescence

40,000 cells were seeded in ibidi six-well flow-chamber slides overnight. The cells were treated with the labeled alpha1-oleate complex (RPMI without serum) for different time periods as mentioned in the figures followed by washing with RPMI (3X) and PBS (2X). The cells were chemically fixed with paraformaldehyde (2% or 4% in PBS). For ER imaging, cells were fixed with 0.1% glutaraldehyde in 3% PFA. The cells

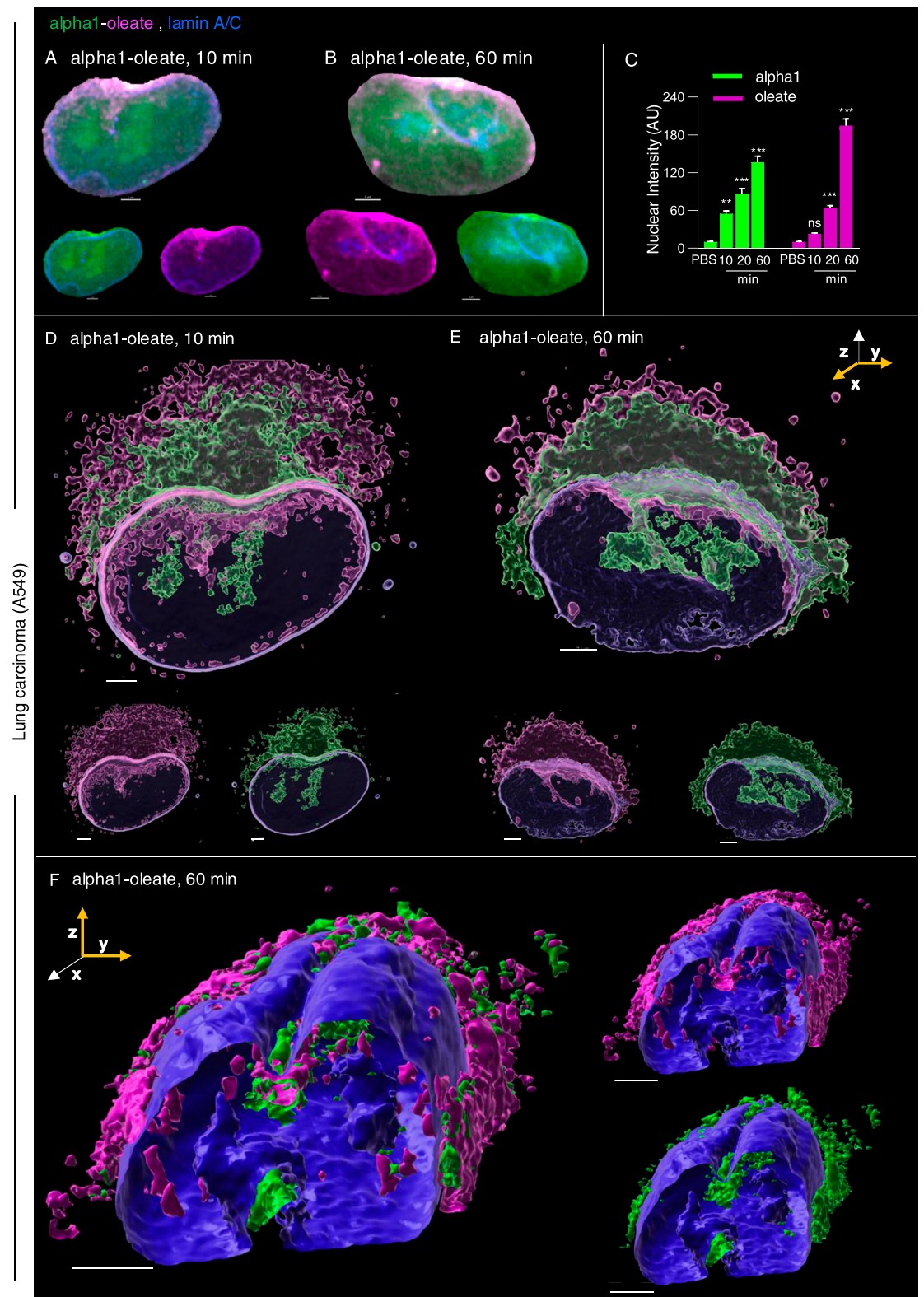

**Figure 5.  Nuclear entry of the alpha1-oleate complex.**
**(A, B)** Nuclear staining of A549 cells exposed to the alpha1-oleate complex. Representative images derived by Airyscan show the nuclear distribution of the labeled alpha1-oleate (35 μM) upon 10 and 60 min of exposure. The JF549-labeled alpha1-peptide (green) and the AF647 click–labeled oleic acid (magenta) were both detected inside the nuclei, after masking the lamin-stained body of the nuclei. Corresponding whole-cell Airyscan fluorescence images used to generate the masked nuclei are provided in Figs 1A and S26. **(C)** Quantification of the time-dependent increase in nuclear uptake of alpha1-oleate. Data are expressed as the mean ± SEM from maximum

were permeabilized (0.3% NP-40, 0.05% Triton X-100, 1X PBS) for 3 min at RT, washed three times in wash buffer (0.05% NP-40, 0.05% Triton X-100, 1X PBS), and treated with Click-iT reagent using Click-iT Cell Reaction Buffer Kit according to the manufacturer's protocol. The clicking reaction was followed by blocking for 1 h with blocking buffer (0.025% Triton X-100, 0.025% NP-40, 5% goat serum, 1X PBS) and incubating with primary (2 h, RT) and secondary (1 h, RT) antibodies as mentioned in the figure legends. Washing steps were performed after incubation with primary and secondary antibodies with wash buffer (3X) for 5 min followed by PBS (3X) and Milli-Q washes (3X). The cells were mounted with ProLong Glass Antifade Mountant and left overnight to dry.

### Live-cell imaging preparation

For live-cell imaging, 80,000 cells were seeded in ibidi 1-well flow-chamber slides overnight. BODIPY-based ER-Tracker Green (1 $\mu$M) was added to the cells in live-cell imaging solution (30 min, 37°C, 5% $CO_2$) followed by washing with live-cell imaging solution (3X). The alpha1-oleate (labeled/unlabeled/mixed labeled complex) was added to the cells and incubated at 37°C, 5% $CO_2$ for time periods mentioned in the figures. The cells were immediately transferred to a confocal microscope (LSM900; Carl Zeiss) for imaging without washing after the incubation periods were over (labeled alpha1-oleate (35 $\mu$M), unlabeled alpha1-oleate (21 $\mu$M), and mixed alpha1-oleate 1:1 vol/vol (labeled, 35 $\mu$M, and unlabeled, 21 $\mu$M)).

For live-cell imaging with halo-KDEL, A549 cells were electroporated using the Amaxa Nucleofector (Nucleofector I; Amaxa Biosystems) as per the manufacturer's instruction (using VCA-1002 protocol) with ~2 $\mu$g of the halo-KDEL plasmid. The nucleofected cells were imaged after 48 h. A typical transfection efficiency of ~60% was observed. Cells expressing the halo-KDEL were incubated with silicon–rhodamine (SiR) with a chloroalkane tag for 30 min at a final concentration of 1 $\mu$M in live-cell imaging solution. The cells were then washed with live-cell imaging buffer and incubated with the unlabeled complex for various time periods as mentioned in figures. The cells were then immediately transferred to a confocal microscope (LSM900; Carl Zeiss) as mentioned above.

PM staining experiments were conducted by first prestaining the cells in a live-cell imaging solution with ER-Tracker for 30 min, followed by staining with CellBrite membrane dye for 15 min. Subsequently, the cells were treated with alpha1-oleate for 10 min.

For live-cell imaging of lipid droplets, cells were grown overnight in ibidi 1-well flow chambers and treated with alpha1-oleate or PBS for the time periods specified in the figures. After treatment, cells were washed three times with PBS and incubated with LipidTOX Deep Red Stain (1:1,000 dilution in live-cell imaging medium) for 30 min at 37°C to allow incorporation into neutral lipid droplets.

For experiments assessing co-localization of lipid droplets with EDVs, lipid droplet staining was performed after silicon–rhodamine staining for Halo-KDEL. After incubation with LipidTOX Deep Red Stain, cells were immediately transferred to a confocal microscope (LSM900; Carl Zeiss) and imaged without washing.

### WGA treatment—pinocytic method

40,000 cells were seeded in ibidi six-well-flow-chamber slides overnight. Cells were treated with WGA-Oregon Green 488 using Influx Pinocytic Cell-Loading Reagent, as per the manufacturer's protocol. After treatment, the cells were recovered in complete media for a minimum of 10 min. The cells were subsequently treated with labeled alpha1-oleate or PBS, followed by click reaction with AF647 for visualization of oleic acid and immunofluorescence with lamin A/C.

### ER stress induction

Stock solutions of ER stress inducers were prepared as follows: tunicamycin (1 mM in DMSO) and thapsigargin (1 mM in DMSO). The working concentrations were 30 $\mu$M for tunicamycin and 1 $\mu$M for thapsigargin. A549 cells were preincubated with the inhibitors for either 1 or 12 h in complete media before live-cell imaging using ER-Tracker. For gene expression and Western blot analysis, cells were treated with ER stress inducers for either 3 or 12 h. In the 3-h experiment, cells were exposed to ER stress inducers for 1 h, followed by an additional 2-h incubation in complete medium (with serum), resulting in a total treatment duration of 3 h.

### Image acquisition and analysis

The confocal images were acquired using laser-scanning confocal microscope (LSM900; Carl Zeiss) equipped with 405-, 488-, 561-, and 647-nm diode lasers for excitation. The 63x/1.40 Oil DIC M27 oil immersion objective lens was used for imaging. For detection of JF-549–labeled alpha-1 and AF647-clicked oleate in the complex, excitation wavelengths of 561 and 647 nm were used. The emission parameters were set to 557–572 nm and 653–668 nm, respectively. For detection of AF488, AF405, and AF 647 secondary labels, excitation wavelengths of 488, 405, and 647 nm were used. The emission parameters were set between 493 and 517, 401 and 422, and 653 and 668 nm, respectively. The four colors were imaged sequentially frame by frame in bidirectional mode. Large field-of-view images using single-plane illumination were captured. 1024 × 1024 images were captured with camera exposures kept below 10 ms. For 3D reconstructions, Z-stacks were acquired, and maximum intensity projections were provided. A Zeiss LSM880 equipped with an

---

intensity projections collected using z-stacks, n = 17 cells for each time point. Statistical significance was determined by the Kruskal–Wallis test with Dunn's multiple comparisons. **P < 0.002, ***P < 0.001, ns, not significant. **(D, E)** Nuclear uptake of the alpha1-oleate complex from the perinuclear area into tubular structures. 3D reconstruction of the entire nucleus and the perinuclear region of an A549 cell exposed to labeled alpha1-oleate (35 $\mu$M) for 10 or 60 min is shown. The nucleus is made transparent to visualize the entry of the complex from the perinuclear compartment. **(F)** Cross-section (y-z) through the 3D renderings of a representative treated A549 cell illustrates the accentuated perinuclear enrichment of the complex and the asymmetrical entry of alpha1-oleate in the nuclear invaginations (mixed complex, 60 min). **(A)** Both constituents were present (see panel (A)), but the process of 3D reconstruction only takes the highest signal into account. In fixed cells, nuclei are visualized by lamin A/C (blue) immunostaining with anti-mouse AF405 secondary antibody. Mixed alpha1-oleate: 1:1 vol/vol, labeled, 35 $\mu$M, and unlabeled, 21 $\mu$M. Scale bars, 2 $\mu$m (A, B, C, D, E, F).

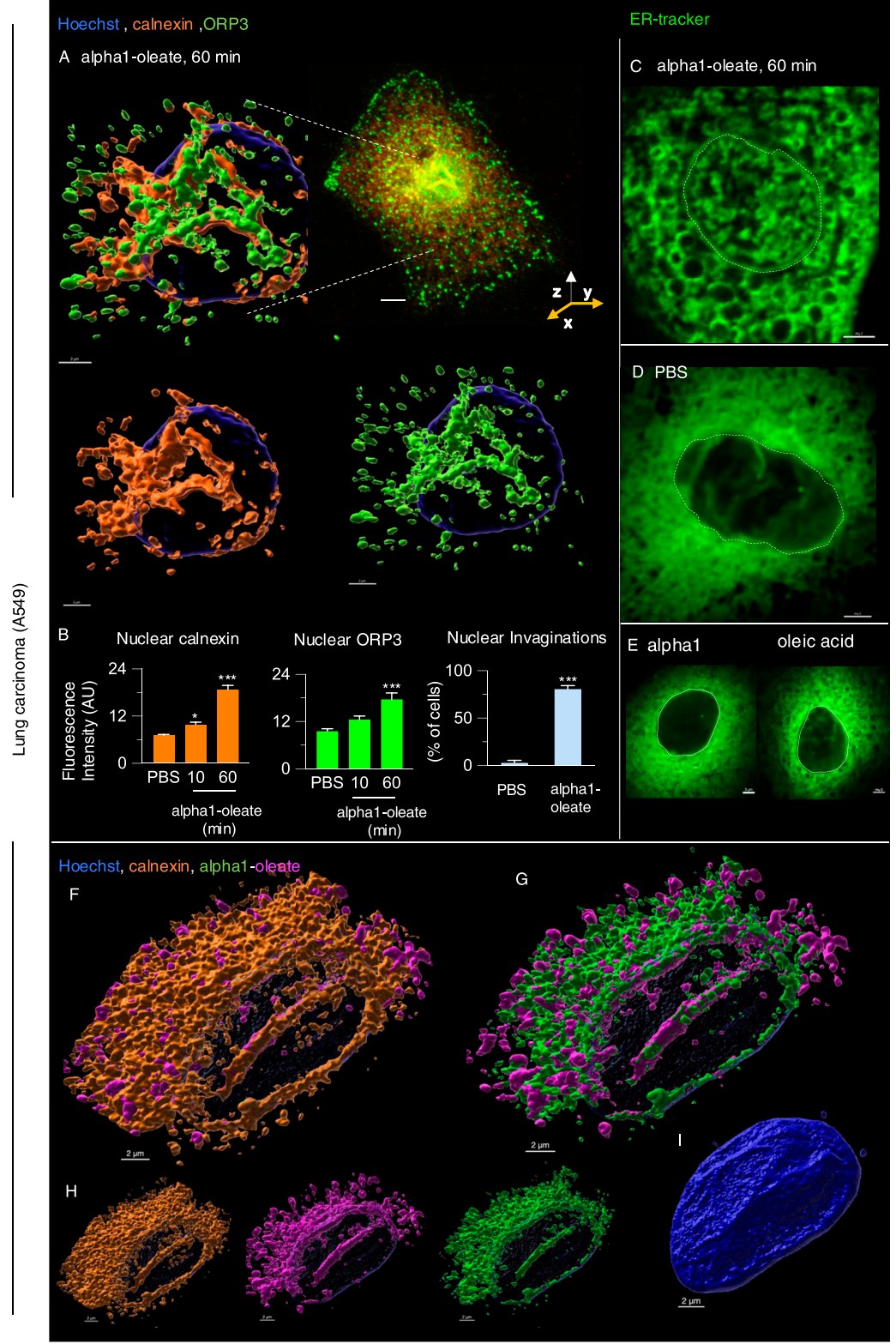

**Figure 6. ER entry inside the nucleus triggered by alpha1-oleate.**
**(A)** 3D reconstruction of the nucleus of an A549 cell exposed to alpha1-oleate showing ER invaginations extending from the perinuclear region into the nuclear interior (unlabeled, 21 μM, 60 min). The ER-resident protein calnexin (cyan) and the ER-interacting protein ORP3 (green) are co-localized with the nuclear invaginations. Calnexin and ORP3 were visualized using secondary anti-rabbit AF647 and anti-mouse AF488, respectively. **(B)** Time-dependent increase in the nuclear content of calnexin and ORP3, and the percentage of cells showing nuclear invagination (observed with calnexin) after alpha1-oleate treatment, quantified from z-stacks. Data are expressed as

Airyscan detector with 100×/1.46 NA oil immersion objective was also used to acquire z-stacks for 3D reconstructions. Airyscan super-resolution (SR) module with 32-channel hexagonal array GaAsP detector was used, and stacks of 40–60 optical sections (0.150 $\mu$m step) were acquired. Airyscan super-resolution image stacks were reconstructed using ZEN 3.1 blue software (Zeiss). The image quantification was done using Fiji/ImageJ (Schindelin et al, 2012) and Imaris (Bitplane). For calculation of total or nuclear uptake using z-stacks, the cell and nuclear surfaces were created. "Sum Intensity" of each channel was calculated with these surfaces. 3D reconstructions were also performed using Imaris.

### Surface reconstructions

3D rendering of the fluorescence images from z-stacks was done using Imaris software (Bitplane, v.9.9) and Imaris viewer (v.9.7.2).

### Concavity analysis for nuclear shape change

The 3D surface of the nucleus was generated from the fluorescence signal of lamin A/C. The Imaris file (format: wrl) was converted to obj format using the software MeshLab 2021. The generated triangle meshworks were imported into MATLAB for further calculation. To reduce calculation time and avoid small-scale curvature measurements, the number of points composing the mesh was decreased and the surface smoothened (http://www.alecjacobson.com/weblog/?p=917, Alec Jacobson script; N_SmoothMesh script, Export Voxel Data, Cyprian Lewandowski, MATLAB file exchange) as described earlier (Biedzinski et al, 2020). The mean curvature was calculated using a script described earlier (Patch Curvature [https://www.mathworks.com/matlabcentral/fileexchange/32573-patch-curvature], MATLAB Central File Exchange. Retrieved 19 December 2021) using the top half of the nucleus to avoid the irregularities in the bottom of the nucleus. To exclude the small unevenness in untreated nuclear surfaces, the concavity percentage with a threshold of −0.1 was calculated for comparison. To measure the degree of invagination, minimum mean curvature was calculated.

### Preparation of GUVs

GUVs were formed by hydrogel-assisted swelling according to established protocols, with modifications previously described (Hansen et al, 2020). Briefly, glass coverslips were sonicated in 1 M NaOH solution (30 min), rinsed in Milli-Q water (3X), and further sonicated (30 min). Coverslips were plasma-etched (1 min) using a BD-20 laboratory corona treater (Electro-Technic Products Inc.) to render the surface clean and hydrophilic. A thin film of 1% (wt/vol) solution of molten ultra-low gelling temperature type IX-A agarose was deposited on the coverslip to provide a reaction bed for GUV formation. The coverslips were placed in AttoFluor cell chambers. Afterward, 25 $\mu$l of lipid mixture which resembles the ER and PM composition in chloroform (25 mg/ml) doped with 4% vol/vol rhodamine C (1 mg/ml) was deposited onto the gelled agarose surface, and the solvent was evaporated with nitrogen gas. The lipid–hydrogel film was rehydrated with 200 mM sucrose in PBS, pH 7.2, for 1 h and then transferred into 200 mM glucose in PBS, pH 7.2, for sedimentation. GUVs were allowed to settle overnight before being seeded on the coverslips for visualization and treatments. The lipid composition used for ER membrane consists of 52.2% egg phosphatidylcholine (PC), 28.5% DOPE, 14.3% phosphatidylinositol (PI), and 5% cholesterol; for PM, the lipid mixture included 29.8% egg-PC, 17.5% DOPE, 3.5% PI, 19.2% sphingomyelin (SM), and 30% cholesterol.

### TUNEL assay

DNA fragmentation was evaluated using the terminal deoxynucleotidyl transferase dUTP nick end-labeling (TUNEL) assay according to the manufacturer's instructions with modifications (Click-iT TUNEL Alexa Fluor 488 imaging assay kit). Cells were fixed (4% PFA, 15 min), permeabilized (DNase-free Proteinase K solution 20 $\mu$g/ml, 15 min), and incubated with TUNEL reaction mixture containing TdT for 60 min at 37°C. After TUNEL reaction, cells were incubated with Click-iT reaction mixture (30 min, 37°C). Cells were counterstained with Hoechst (10 $\mu$M in PBS, 10 min), mounted in Fluoromount aqueous mounting media, and imaged using a microscope (Zeiss). Fluorescence intensities were quantified by ImageJ. The net mean fluorescence intensity was calculated after subtraction of background fluorescence.

### Western blot

350,000 cells were grown overnight in six-well plates. The cells were washed with RPMI without serum. Alpha1-oleate– or PBS-treated cells (60 min) were lysed with Pierce RIPA Buffer supplemented with protease inhibitor (25x) and phosphatase inhibitor (10X). The cell lysate was freeze-thawed using dry ice at room temperature five times and centrifuged at 16,100$g$ for 20 min in 4°C, and the supernatant was collected. For cell fractionation, $2 \times 10^6$ cells were seeded in 10-cm culture dishes overnight. The fractions were separated using Qproteome Cell Compartment Kit. Protein concentration was measured using Pierce 660 nm Protein Assay Reagent according to the manufacturer's instructions. Cell lysates were run in a 4–16% Bis-Tris

the mean ± SEM of three independent experiments, n = 15 cells. Statistical significance was determined by the Kruskal–Wallis test with Dunn's multiple comparisons (for nuclear calnexin); mean ± SEM of two independent experiments, n = 15 cells, one-way ANOVA with Šidák's multiple comparison tests (for nuclear ORP3); mean ± SEM of three independent experiments, n = 50 cells at least, two-tailed unpaired $t$ test (for nuclear invaginations). ***$P < 0.001$, *$P < 0.033$. **(C, D)** Accentuated ER staining inside the nucleus using ER-Tracker (green) in alpha1-oleate (unlabeled, 21 $\mu$M, 60 min)–treated A549 cell compared with the control (PBS). Representative live-cell confocal images. **(E)** Control experiments in A549 cells exposed to alpha1-peptide (21 $\mu$M) or oleic acid (105 $\mu$M) show no nuclear ER staining. **(F, G)** A549 cells were exposed to the alpha1-oleate complex formed by JF549-labeled alpha1-peptide (green) and the AF647 click–labeled oleic acid (magenta). 3D reconstruction of a transparent nucleus from a representative A549 cell treated with alpha1-oleate (mixed complex, 60 min) shows the presence of both constituents in the nuclear invaginations lined by the ER (calnexin). Calnexin and ORP3 were visualized using secondary anti-rabbit AF647 and anti-mouse AF488, respectively. **(H)** Individual channels are shown. **(I)** Solid body of the nucleus is shown, suggesting nuclear shape change. The corresponding fluorescence image of the whole cell is provided in Fig S26. Mixed alpha1-oleate: 1:1 vol/vol, labeled, 35 $\mu$M, and unlabeled, 21 $\mu$M. Scale bar, 2 $\mu$m (A, F, G, H, I); 3 $\mu$m (C, D, E); 5 $\mu$m ((A), whole cell).

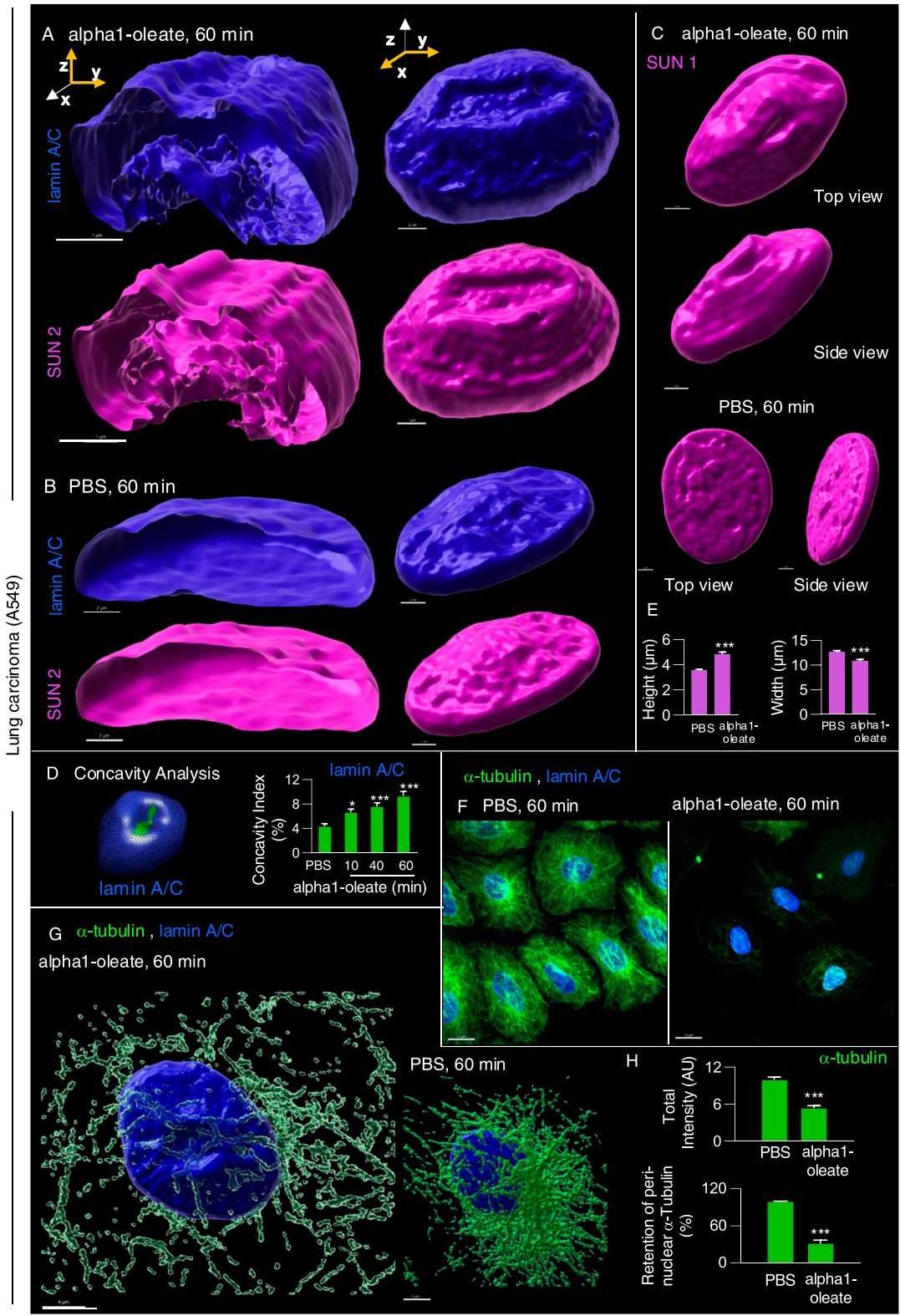

**Figure 7. Nuclear shape change investigated by staining for inner nuclear membrane constituents and microtubular network.**
**(A)** Nuclear shape change, defined by staining the lamin nucleoskeleton (blue) and inner nuclear membrane protein SUN2 (magenta) fluorescence signals and visualized by 3D reconstructions of the nucleus in alpha1-oleate (unlabeled, 21 µM)–treated A549 cells. **(B)** Controls of nuclear shape from cells exposed to PBS. The corresponding y-z cross-sections (left) illustrate the transition from a smooth, rounded nuclear shape to a deformed morphology with large invaginations. **(C)** Shape change of the nucleus of an A549 cell treated with alpha1-oleate (unlabeled, 21 µM) for 60 min, visualized with another inner nuclear membrane marker SUN1 (magenta).

gel under denaturing conditions and transferred to Trans-Blot Turbo Mini PVDF Membrane using Trans-Blot Turbo Transfer System (Bio-Rad). Primary antibodies were detected with horseradish peroxidase–conjugated (HRP-conjugated) secondary anti-mouse/rabbit antibody. The immunoblots were treated with respective primary antibodies overnight at 4°C, followed by secondary antibody for 1 h at RT. The blots were visualized using Novex ECL substrate in ChemiDoc XRS+ (Bio-Rad).

### In vitro transcriptomic analysis

A549 cells were seeded in six-well tissue culture plates (35,000 cells/well in 2 ml in RPMI/5% FBS), allowed to adhere overnight, washed with RPMI without serum, and treated with alpha1-oleate (35 $\mu$M) or PBS, in RPMI (1 h, at 37°C in 5% CO2). Alternatively, FBS (5%) was added after 1 h and cells were incubated for 2 more hours (3-h time point). Total RNA was extracted from A549 cells using RNeasy Mini Kit (QIAGEN) and on-column DNase digestion. 100 ng of RNA was amplified using GeneChip 3′IVT PLUS Kit (Thermo Fisher Scientific), then fragmented, and labeled RNA was hybridized onto Human Genome U219 array strips (16 h at 45°C), washed, stained, and scanned in-house using the GeneAtlas system (Affymetrix). All samples passed the internal quality controls included in the array strips (signal intensity by signal-to-noise ratio; hybridization and labeling controls; sample quality by GAPDH signal; and 3′-5′ ratio < 3).

Data were normalized using Robust Multi-Average implemented in Transcriptome Analysis Console (v.4.0.1.36, Applied Biosystems) software. Differential expression was computed by comparing treated cells with PBS control. Differentially expressed probes were sorted by relative expression (two-way ANOVA model using method of moments), and an absolute fold change > 1.5 was considered significant (Eisenhart, 1947). Heatmaps were constructed using Prism software. Significantly altered genes and regulated pathways were analyzed using Ingenuity Pathway Analysis software (IPA, Ingenuity Systems; QIAGEN).

### Statistical analysis

Data are expressed as the means ± SEM. The normality of the data distribution was determined by the Shapiro–Wilk normality test. The different statistical analyses performed using GraphPad Prism 9 are mentioned in the figure legends. For kinetic studies, statistical significance was determined by one-way ANOVA with

Šidák's multiple comparison test. Statistical significance for parametric analysis was determined by unpaired t-tests or one-way ANOVA with Šidák's multiple comparison test for kinetic studies, whereas for nonparametric analysis, the Kruskal–Wallis or Mann–Whitney test with Dunn's multiple comparisons was used.

## Data Availability

This study includes no data deposited in external repositories. All data generated or analyzed during this study are included in the article.

## Supplementary Information

## Acknowledgements

We acknowledge Maria Trulsson (Zeiss Research Microscopy Solutions) and Salla Marttila (Swedish University for Agricultural Sciences) help for Airyscan imaging. We thank Prof. Gražvydas Lukinavičius (Max Planck Institute for Biophysical Chemistry) for the gift of HaloTag SiR. We acknowledge the help of Michelle Cavalera and Parisa Esmaeili for technical help with experiments. This study was funded by grants from the Swedish Cancer Society (Cancerfonden), Swedish Research Council (Vetenskapsrådet), Åke Wibergs Foundation, HAMLET BioPharma AB, Lund, Sweden, and the Royal Physiographic Society in Lund. Support to the Svanborg group was further provided from the European Union's Horizon 2020 research and innovation program under grant agreement No. 954360.

### Author Contributions

S Sabari: data curation, formal analysis, validation, investigation, and writing—review and editing.
S Chinchankar: data curation, formal analysis, and writing—review and editing.
I Ambite: data curation, formal analysis, and methodology.
A Nazari: data curation and formal analysis.
APNBM Carneiro: data curation and formal analysis.
A Svenningsson: data curation and formal analysis.

The SUN1/SUN2 and lamin A/C were visualized using secondary anti-rabbit AF647 and anti-mouse AF405 antibodies, respectively. **(D)** Nuclear shape change was quantified using concavity analysis, which detected an increase in concavity over time. The concave portion of a representative alpha1-oleate–treated nucleus (blue) generated using MATLAB from z-stacks is represented with a green color. Statistical significance was determined by the Kruskal–Wallis test with Dunnett's multiple comparison test, n = 50 cells. *$P$ < 0.033, ***$P$ < 0.001. **(E)** Increase in height and decrease in width of nucleus in alpha1-oleate–treated cells were quantified. Statistical significance was determined by mean ± SEM, Mann–Whitney two-tailed analysis, n = 40 cells. ***$P$ < 0.001. **(F)** Loss of $\alpha$-tubulin staining in alpha1-oleate–treated cells (unlabeled, 21 $\mu$M) for 60 min compared with control. **(G)** Loss of the dense perinuclear microtubular network and disruption of the remaining filaments with alpha1-oleate treatment. **(G)** Representative 3D reconstructions shown in (G). The $\alpha$-tubulin and lamin A/C were visualized using secondary anti-mouse AF488 and anti-rabbit AF647 antibodies, respectively. **(H)** Quantifications of $\alpha$-tubulin fluorescence intensity and perinuclear density of $\alpha$-tubulin. Statistical significance was determined by the mean ± SEM, Mann–Whitney two-tailed analysis, n = 45 cells (total $\alpha$-tubulin intensity). Data are expressed as the mean ± SEM of three independent experiments, n = 50 cells. Statistical significance was determined by a two-tailed unpaired $t$ test (loss of perinuclear $\alpha$-tubulin). ***$P$ < 0.001. Scale bar, 2 $\mu$m (A, B, D); 15 $\mu$m (F); 5 $\mu$m (G).

C Svanborg: conceptualization, resources, supervision, funding acquisition, project administration, and writing—original draft, review, and editing.

A Chaudhuri: conceptualization, data curation, formal analysis, supervision, funding acquisition, investigation, methodology, and writing—original draft, review, and editing.

## Conflict of Interest Statement

C Svanborg and I Ambite hold shares in HAMLET BioPharma, as a representative of scientists in the HAMLET group. Patents protecting the use of the alpha1-oleate complex have been granted. Other authors declare no conflict of interest.

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
