## [Reviewer comments · Life Science Alliance]

Life Science Alliance

Rapid ER remodeling induced by a peptide-lipid complex in dying tumor cells

Samudra Sabari, Siddharth Chinchankar, Ines Ambite, Atefeh Nazari, António Carneiro, Axel Svenningsson, Catharina Svanborg, and Arunima Chaudhuri

DOI: <https://doi.org/10.26508/lsa.202403114>

Corresponding author(s): Catharina Svanborg, Lund University

Review Timeline:

Submission Date:	2024-10-28
Editorial Decision:	2024-12-02
Revision Received:	2025-03-05
Editorial Decision:	2025-03-07
Revision Received:	2025-03-12
Accepted:	2025-03-13

Transaction Report:

December 2, 2024

Re: Life Science Alliance manuscript #LSA-2024-03114-T

Prof. Catharina Svanborg
Lund University
Laboratory Medicine, MIG
Klinikgatan 28
BMC B13
Lund, Skane 22242
Sweden

Dear Dr. Svanborg,

Thank you for submitting your manuscript entitled "Remodeling and nuclear entry of ER by peptide-lipid complex packages dying cell contents in nucleus" to Life Science Alliance. The manuscript was assessed by expert reviewers, whose comments are appended to this letter. We invite you to submit a revised manuscript addressing the Reviewer comments.

Thank you for this interesting contribution to Life Science Alliance. We are looking forward to receiving your revised manuscript.

Sincerely,

B. MANUSCRIPT ORGANIZATION AND FORMATTING:

Reviewer #1 (Comments to the Authors (Required)):

The alpha1-oleate has been shown to exhibit tumoricidal effect. In this manuscript, the authors monitor the intracellular impact upon alpha1-oleate treatment. They found that the complex has drastic ER remodeling activity, including loss of peripheral network, vesiculation and fragmentation. In addition, the nuclear envelope deformation was observed with nuclear entry of other intracellular structure. Overall, these findings represent an interesting membrane remodeling case that comes with cancer cell killing effect. Imaging analysis was nicely performed. However, the underlying mechanism is largely unknown at this stage and the significance of nuclear entry of organelles are not clear.

Major points:

1. It is noticeable that in treated A498 cells, there are quite some bright puncta. Are these EDVs as shown later or something else?

2. In the GUV assays, membranes are visualized but the complex is not. It is intriguing whether the alpha1-oleate is mostly incorporated into membranes. Would they enter the GUV or even lyse the membrane? These parameters need to be measured in the assay.

Similar concerns are with the in-cell experiments. The complex can apparently penetrate plasma membrane. Would it cause leakage of the PM? The authors explain the complex to prefer ER membranes than the PM due to difference in lipid composition. But how then it gets into the cell?

3. The description starting from figure 5 seems less informative. The nuclear entry part does not reflect an active process, but rather a consequence of ER destruction. It is less likely that cytoplasmic components are delivered there on purpose. Please be cautious about the wording there, in the title and abstract.

Minor point:

Sth is wrong with the title of figure 6.

Reviewer #2 (Comments to the Authors (Required)):

1. Short summary of the paper:

This manuscript by Sabari et al. provides an extensive morphological characterization of the impact of alpha1-oleate on the ER network and nuclear shape and reports on the exciting formation of an "ER-nuclear compartment", where ER enters the nucleus. The authors provide support for the efficient uptake of the alpha1-oleate complex into different carcinoma cell lines as well as co-localization of the constituents at the plasma membrane and in particular in perinuclear regions. In a GUV model, alpha1-oleate preferentially deformed vesicles representative of ER membrane lipid composition compared to plasma membrane. Treatment with alpha1-oleate leads to the formation of numerous ER-derived vesicles and ER stress, which induces a branch of the unfolded protein response as well as apoptosis and a transcriptional response that includes the inhibition of cancer-related gene expression. The entry of ER into the nucleus and the deformation of the nucleus is accompanied by a disruption of the microtubule network. In addition to ER, also Golgi and ribosomes (60S visualized via RPL3) have been found to enter the nucleus upon alpha1-oleate treatment. The authors suggest that alpha1-oleate treated dying cells package cellular material into the nucleus to withhold the cargo until the cells are removed from the tumor tissue.

2. Specific comments:

In general, the manuscript is well written and the data is of high quality and large volume. Though this is a purely morphological description of cellular changes triggered by alpha1-oleate without new mechanistic insights, this study still provides interesting new findings that will likely be of interest and will serve as basis for the further elucidation of the effects of alpha1-oleate on membranes of the ER and the continuous nuclear envelope. The conclusions are well-supported by the data.

I have only a few concerns that should be addressed:

i)

The authors find entry of ER into the nucleus upon alpha1-oleate treatment (visualized by different means, e.g. ER-tracker, calnexin and ORP3). At the same time, they present data demonstrating that the alpha1-oleate complex enters the nucleus. However, the authors do not assess whether these ER structures co-localize with nuclear alpha1-oleate. Do these structures overlap? Does ER penetrate the nucleus at sites where the alpha1-oleate complex is entering as well?

e.g. for the Golgi entering the nucleus, the authors have the respective micrographs already at hand: in Appendix Figure S31C, the authors use alpha1-oleate (both labeled to follow complex and single constituents) and then assess Golgi localization inside the nucleus. Still, they do not show the respective panels that would allow an evaluation of whether the Golgi membranes that move into the nucleus actually co-localize with alpha1-oleate. Could the authors please show respective overlay panels?

ii)

Information missing for Figure 2: Could the authors please indicate how they have measured the distance between the plasma membrane and the ER as quantified in Figure 2E "Distance from plasma membrane"? Does this correspond to the pictures shown in 2G and H? As in these pictures, the CellBrite stained PM in PBS control seems to be more distant to the ER than in alpha1-oleate treated cells, though quantifications shown say otherwise. This quantification should be explained and also indicated in the corresponding pictures for clarity.

iii)

The formation of ER-derived vesicles (EDVs) is really interesting, and the authors observed these EDVs using ER-tracker, calnexin or Halo-KDEL (though it remains unclear whether these are actually the same structures or distinct ones). The ER-tracker-marked EDVs seem to contain labeled oleate (Fig. 3A-D and also Appendix Fig. S8), but it remains unclear whether they also contain alpha1. In addition: Can the authors exclude that those ER-tracker surrounded structures correspond to lipid droplets that emerge from the ER and remain coated with ER-tracker and accumulate labeled oleate?

In the same line: can the authors exclude that the formation of EDVs and general membrane deformation is simply due to high amounts of oleate, which has been already shown to induce NER proliferation? As a control, the authors use PBS throughout the paper. In Figure 1 they show that the labeled single constituents of the alpha1-oleate complex do not enter individually. However, unlabeled oleate (as used for several of the experiments shown) can of course enter the cells. An additional control using unlabeled oleate (not in complex with alpha1) in comparable concentrations is necessary to exclude any effect of oleate alone (e.g. for the data shown in Figure 3).

3. Minor comments in respect to text changes and data presentation:

2. Figure 1: Could the authors comment on the pronounced foci formation of the complex in kidney carcinoma (A498) cells? Alpha1 and oleate seem to perfectly co-localize at these multiple punctate structures that only seem to exist in this specific cell line shown in Figure 1H (not in the other cell lines shown), but nothing in this respect is mentioned in the text.

No complete (or rather very little) overlap between the single constituents (e.g. 5A) in the nucleus seems to exist. The authors should mention that mostly the single constituents can be detected, and that only limited overlap exists between the single constituents alpha1 and oleate in the nucleus in Figure 5A after 10 min of incubation with alpha1-oleate in the nucleus.

The title "Remodeling and nuclear entry of ER by peptide-lipid complex packages dying cell contents in nucleus" is quite hard to understand, maybe the authors could consider rephrasing?

Page 3: Seem that either domain or peptide is too much in the following sentence:
"The N-terminal alpha-helical domain peptide of alpha-lactalbumin"

Appendix Fig S2: The panels A-C correspond to different labeling times, but what are the three different micrographs shown for A, for B and for C? The legend does not explain.

There is one yellow-highlighted reference in the legend for Appendix Fig S4 left.

Information in Appendix Fig. S5 and in corresponding legend is missing in respect to the cell line shown here.

The authors do not mention or discuss Fig. 2F in the text when Figure 2 is described, but describe their finding from Fig. 2F at the end of the text for Figure 3. This is confusing, please adapt either text or figure panel sequence. 2F seems to rather belong to Figure 3 (shows in principle the same as Figure 3A and B).

The organization of Figure 3 seems a bit confusing: why is the single channel for ER-tracker shown as panel 3A, the corresponding overlay as 3B and the corresponding alpha1-oleate channel in magenta in between without panel descriptor? Should this not all be 3A?

Figure 5: Why is the 3D model of the same cell shown in Fig. 5A and Appendix Fig 16 B (model for overlay as well as single channels combined with nucleus). Similarly, Appendix Fig. 18B shows the same 3D model of a cell as shown in 5H.

Figure 4K and L: mostly not readable. What does the colour code mean (grey, different shades of yellow, magenta, pink, blue)? Similarly, gene names in 4N not readable, partially due to names overlapping. No use for gene names if they are too small to read.

Reviewer #3 (Comments to the Authors (Required)):

In this manuscript, authors show that the alpha1-oleate treatment significantly changes the shape of the nucleus, the ER, and microtubule network. Their data for this part are convincing. However, it is not convincing at all whether the ER, ribosomes, and Golgi enter the nuclei of alpha1-oleate treated cells. While this study has a potential to provide mechanistic insights into how alpha1-oleate enters nucleus and potentially changes the transcriptome, solid evidence is lacking. In my opinion this study requires substantial revisions to be published in LSA. I hope that the authors find the detailed comments below useful in revising their manuscript.

- specific major concerns essential to be addressed to support the conclusions

(i) Lack of solid evidence for the nuclear entry of the alpha1-oleate, ER, ribosomes, and Golgi.

In Figure 5, while the authors show that alpha1-oleate signal is "inside" the nucleus in the 3D surface reconstruction, it looks like an artifact due to light scattering. In other words, because there are bright fluorescent signal at the nuclear periphery (outside of nucleus), some of the signal seems to be inside the nucleus even though it actually is not. The same applies to Figures 6 and 8, in which ER proteins, ribosome subunits, and Golgi markers appear to be enriched at the nucleoplasmic reticulum (NER). Even though it is very interesting that the ER, ribosome, and Golgi proteins accumulate at the NER, judging from the images they seem to be outside the nucleus and the signal inside the nucleus is just scattered light from the bright perinuclear signals. If authors want to claim that they are inside the nucleus, they need to provide much more convincing data, e.g., by performing electron microscopy and show if the ER, ribosomes, and Golgi are indeed inside the nucleus.

(ii) Lack of control experiments to show whether only the alpha1-oleate complex can induce such morphology change or the alpha1 peptide or oleic acid alone can do.

Even though the authors show that the alpha1 peptide or oleic acid alone do not enter the cell as much as the alpha1-oleate complex does in Figure 1, such control experiments are not performed for the following experiments. Because it is possible that the peptide or oleic acid alone can trigger some signaling pathways that cause morphology changes of the nucleus, the ER, and microtubule network, the authors should perform those experiments using the peptide or oleic acid alone.

- minor concerns that should be addressed

Fig. 1: The authors claimed that "the alpha1 peptide and oleic acid were both detected in the cytoplasm and nuclei". It is hard to judge this from the projection images that they show. They should show single confocal slice images. In addition, they need to quantify the alpha1-oleate signal in the cytoplasm and nucleoplasm separately in 2D slice images instead of measuring signal in the whole cells in the projected image.

Fig. 2F: It is hard to see the alpha1-oleate signal from this merged image and judge the colocalization with the ER marker. They should show images of individual channels.

Appendix Fig. 13: Atlastin and Reticulon localization do not agree with their known/reported localization at the ER. The authors should find alternative ways to visualize those proteins, or simply remove those data.

Reviewer #1 (Comments to the Authors (Required)):

The alpha1-oleate has been shown to exhibit tumoricidal effect. In this manuscript, the authors monitor the intracellular impact upon alpha1-oleate treatment. They found that the complex has drastic ER remodeling activity, including loss of peripheral network, vesiculation and fragmentation. In addition, the nuclear envelope deformation was observed with nuclear entry of other intracellular structure. Overall, these findings represent an interesting membrane remodeling case that comes with cancer cell killing effect. Imaging analysis was nicely performed. However, the underlying mechanism is largely unknown at this stage and the significance of nuclear entry of organelles are not clear.

We thank the reviewer for these comments.

Major points:

1. It is noticeable that in treated A498 cells, there are quite some bright puncta. Are these EDVs as shown later or something else?

Fig. 1 shows the uptake of the complex defined by staining of both constituents, which has not previously been demonstrated.

In response to the reviewer's comment, we have re-examined images of the different cell types captured by Airyscan. The formation of puncta/vesicles was observed in the cytoplasm and perinuclear areas of most cells, consistent with the pattern observed in A498 cells. We have therefore replaced the panels in Fig.1 C, H, I, with cells where such punctate staining was observed (60 min exposure). Panels A and B show A549 cells exposed to alpha1-oleate for 10 minutes, where puncta or vesiculation was less prominent.

Subsequent experiments show that many of the bright puncta are ER membrane lined and contain the complex. Fig. 3 shows the presence of the complex in the membrane lining the EDVs and inside the EDVs in A549 cells. The Results and Figure legends have been revised accordingly.

Data addressing the difference between membrane-lined ER-derived vesicles and lipid droplets have also been added to Fig.3. See also the response to Reviewer 2.

2. In the GUV assays, membranes are visualized but the complex is not. It is

intriguing whether the alpha1-oleate is mostly incorporated into membranes. Would they enter the GUV or even lyse the membrane? These parameters need to be measure in the assay.

Similar concerns are with the in-cell experiments. The complex can apparently penetrate plasma membrane. Would it cause leakage of the PM? The authors explain the complex to prefer ER membranes than the PM due to difference in lipid composition. *But how then it gets into the cell?*

Most of these interesting and important questions have been addressed in previous publications and more extensive comments have been added in the revised manuscript.

a. HAMLET and alpha1-oleate triggers rapid membrane blebbing and tubulation in lipid bilayers in the absence of other membrane constituents or energy sources (Nadeem et al. (2015) Sci Rep. 12:5:16432; Hansen et al. (2020) Mol Biol Evol 37: 3083-3093). Alpha1-oleate has further been shown to induce GUV division, which included labeled dextran cargo and was discussed as a model for protocell division (Hansen et al. (2020) Mol Biol Evol 37: 3083-3093).

b. Alpha1-oleate and HAMLET trigger rapid ion fluxes across cellular membranes (Storm et al. (2013) PLoS ONE 8(3): e58578; Nadeem et al. (2015) Sci Rep. 12:5:16432; Brisuda et al. (2021) Nat Commun 12: 3427; Tran et al. (2023) Int J Cancer. 154:1–16). Uptake of the complex has not been detected in the GUV model.

c. Earlier studies have demonstrated the integration of alpha1–oleate complexes into vesicle membranes (Hansen et al. (2020) Mol Biol Evol 37: 3083-3093). Alpha1-oleate complexes formed by the AF488-labeled peptide showed strong colocalization with the rhodamine membrane marker within 3–5 minutes of exposure. Integration of the peptide constituent of the complex was visualized in model membranes. Membrane integration of the click labeled oleic acid cannot be detected in the GUVs, as the click technology requires fixation and addition of reagents post fixation.

d. This study shows that the ER membrane lipid composition favors the membrane interaction of the complex, which we interpret as a mechanism underlying the ER selectivity of the complex that is shown here. The exact mechanism of membrane integration of the complex is not yet known, but the pattern is not suggestive of leakage or pore formation in the GUV model (Nadeem et al. (2015) Sci Rep. 12:5:16432). Previous studies have investigated the effect of membrane lipid composition on the response to alpha1-oleate (Ho et al., 2022, BioFactors 48: 1145–1159). For example, the addition of cholesterol has been shown to reduce the membrane response to the alpha1-oleate complex (Ho et al., 2022, BioFactors 48: 1145–1159).

3. The description starting from figure 5 seems less informative. The nuclear entry part does not reflect an active process, but rather a consequence of ER destruction. It is

less likely that cytoplasmic components are delivered there on purpose. Please be cautious about the wording there, in the title and abstract.

The paper has been extensively revised to focus on the change in ER structure induced by alpha1-oleate. The ER network is remodeled but not destroyed. The presence of the remaining ER is clearly demonstrated in the perinuclear and nuclear areas, as well as its content of alpha1-oleate.

The gene expression analysis clearly identifies an active apoptotic response as well as an ER stress response in the cells and a partial ER stress response is identified at the protein level, involving only eIF2 α (PERK pathway). We therefore suggest that this is an active process and discuss it as a novel ER response mechanism.

The reviewer further raises the question if the nuclear accumulation reflects an active process or is a consequence of ER destruction. We assume that a destroyed ER would be fragmented and degraded in the cytoplasm. Why would a destroyed ER interact specifically with the nuclear membrane constituents and cause this stepwise change in nuclear shape and content?

To compare the effects of alpha1-oleate to previously known UPR inducing compounds, we now include data obtained from cells treated with thapsigargin, or tunicamycin. There was no evidence of ER vesiculation, perinuclear accumulation or increased nuclear entry of the ER in Thapsigargin or Tunicamycin treated cells (Fig. 8).

Minor point:

Sth is wrong with the title of figure 6.

Has been corrected.

Reviewer #2 (Comments to the Authors (Required)):

1. Short summary of the paper:

This manuscript by Sabari et al. provides an extensive morphological characterization of the impact of alpha1-oleate on the ER network and nuclear shape and reports on the exciting formation of an "ER-nuclear compartment", where ER enters the nucleus. The authors provide support for the efficient uptake of the

alpha1-oleate complex into different carcinoma cell lines as well as co-localization of the constituents at the plasma membrane and in particular in perinuclear regions. In a GUV model, alpha1-oleate preferentially deformed vesicles representative of ER membrane lipid composition compared to plasma membrane. Treatment with alpha1-oleate leads to the formation of numerous ER-derived vesicles and ER stress, which induces a branch of the unfolded protein response as well as apoptosis and a transcriptional response that includes the inhibition of cancer-related gene expression. The entry of ER into the nucleus and the deformation of the nucleus is accompanied by a disruption of the microtubule network. In addition to ER, also Golgi and ribosomes (60S visualized via RPL3) have been found to enter the nucleus upon alpha1-oleate treatment. The authors suggest that alpha1-oleate treated dying cells package cellular material into the nucleus to withhold the cargo until the cells are removed from the tumor tissue.

We thank the reviewer for this summary.

2. Specific comments:

In general, the manuscript is well written and the data is of high quality and large volume. Though this is a purely morphological description of cellular changes triggered by alpha1-oleate without new mechanistic insights, this study still provides interesting new findings that will likely be of interest and will serve as basis for the further elucidation of the effects of alpha1-oleate on membranes of the ER and the continuous nuclear envelope. The conclusions are well-supported by the data.

We thank the reviewer for these positive comments

I have only a few concerns that should be addressed:

i)

The authors find entry of ER into the nucleus upon alpha1-oleate treatment (visualized by different means, e.g. ER-tracker, calnexin and ORP3). At the same time, they present data demonstrating that the alpha1-oleate complex enters the nucleus. However, the authors do not assess whether these ER structures co-localize with nuclear alpha1-oleate. Do these structures overlap? Does ER penetrate the nucleus at sites where the alpha1-oleate complex is entering as well?

This is a very interesting question, which has been addressed by additional experiments in the revised manuscript (Fig.5C-F). The text of the results section has been revised accordingly.

“These results suggest that there are at least two mechanisms leading to nuclear entry of the complex. One is detected as diffuse staining of the peptide and oleic acid, symmetrical and apparently unrelated to NER formation. The second phase of entry is marked by an increase in NER formation. The complex enters the nucleus encapsulated within the NER, but it remains unclear whether it ultimately reaches the nuclear lumen or remains enclosed within the ER as nuclear changes progress. Nonetheless, the NERs and complex are clearly shown to be located inside the nuclei.”

e.g. for the Golgi entering the nucleus, the authors have the respective micrographs already at hand: in Appendix Figure S31C, the authors use alpha1-oleate (both labeled to follow complex and single constituents) and then assess Golgi localization inside the nucleus. Still, they do not show the respective panels that would allow an evaluation of whether the Golgi membranes that move into the nucleus actually co-localize with alpha1-oleate. Could the authors please show respective overlay panels?

The revised paper focuses on the ER response rather than the fate of the ER associated organelles including the Golgi.

ii)

Information missing for Figure 2: Could the authors please indicate how they have measured the distance between the plasma membrane and the ER as quantified in Figure 2E "Distance form plasma membrane"? Does this correspond to the pictures shown in 2G and H?

The distance between the plasma membrane and the retracting ER network was quantified at 63X magnification, by drawing lines between the edge of the retracting

ER network, defined by ER tracker staining, and the edge of the cell, defined by brightfield microscopy. This information has been added to the revised manuscript.

As in these pictures, the CellBrite stained PM in PBS control seems to be more distant to the ER than in alpha1-oleate treated cells, though quantifications shown say otherwise. This quantification should be explained and also indicated in the corresponding pictures for clarity.

The loss of a structurally defined peripheral ER network was seen in 100% of the cells.

CellBrite staining was introduced to visualize the plasma membrane but CellBrite labeled a wider area in the cell periphery, which also included the peripheral ER. After alpha1-oleate treatment, this wider staining at the cell periphery was lost, leaving a more linear staining at the border of the cell, interpreted as the plasma membrane and a retracting and structurally chaotic ER further in. This has been clarified in the revised manuscript.

iii)

The formation of ER-derived vesicles (EDVs) is really interesting, and the authors observed these EDVs using ER-tracker, calnexin or Halo-KDEL (though it remains unclear whether these are actually the same structures or distinct ones). The ER-tracker-marked EDVs seem to contain labeled oleate (Fig. 3A-D and also Appendix Fig. S8), but it remains unclear whether they also contain alpha1. In addition: Can the authors exclude that those ER-tracker surrounded structures correspond to lipid droplets that emerge from the ER and remain coated with ER-tracker and accumulate labeled oleate?

We thank the reviewer for these interesting questions.

i) Lipid droplets

The question about lipid droplets was addressed experimentally in cells transfected with the Halo-KDEL ER marker. Lipid droplets were stained using LipidTOX™ Red.

In untreated A549 cells, LipidTOX™ Red staining revealed the presence of lipid droplets throughout the cells, including the cell periphery and the perinuclear area. Cells treated with alpha1-oleate showed a similar distribution of lipid droplets, with an evidence of an increase in number (Fig. S13).

The ER derived vesicles (EDVs) formed in response to alpha1-oleate treatment were identified in cells transfected with the halo-KDEL plasmid, which labels the ER structures. Cells were subsequently co-stained with silicon Rhodamine dye (for ER visualization via halo-KDEL) and LipidTOX™ Red Neutral Lipid Stain to label LDs. This dual staining allowed for the identification of the lipid droplets and EDVs and their spatial relationship.

Interestingly, the majority of EDVs stained by halo-KDEL were not stained by LipidTOX™ suggesting that the EDVs represent a separate class of vesicular structures (Fig. 3M-O, Fig. S14). This information has been added to the revised manuscript.

In the same line: can the authors exclude that the formation of EDVs and general membrane deformation is simply due to high amounts of oleate, which has been already shown to induce NER proliferation? As a control, the authors use PBS throughout the paper. In Figure 1 they show that the labeled single constituents of the alpha1-oleate complex do not enter individually. However, unlabeled oleate (as used for several of the experiments shown) can of course enter the cells. An additional control using unlabeled oleate (not in complex with alpha1) in comparable concentrations is necessary to exclude any effect of oleate alone (e.g. for the data shown in Figure 3).

We are aware of the interesting paper by Ridgway et al. showing oleic acid induced NE tubulation where they use ConA as cellular cargo.

In control experiments, we found little evidence of EDV formation or an increase in lipid droplet formation in cells exposed to oleate alone. We also did not observe any changes in the nuclear ER at the concentration of oleic acid used here (Figs.S8 and S31).

We further performed experiments with increased oleic acid concentrations (500 uM, Lagace & Ridgway (2005) Mol Biol Cell. 16:1120–1130). There was an increase in

nuclear invaginations, confirming the published data (Fig. S35). The extent of tubulation was lower than in cells exposed to the complex with 105 μ M of oleate and there was no evidence of EDV formation.

The revised paper discusses how the presentation of high amounts of oleic acid to the nuclear membrane, may occur as a result of perinuclear accumulation. The data also suggests that high amounts of oleic acid may contribute to the effects on the nuclear membrane, which are characterized in this study.

These observations do not exclude the effects of oleate alone but illustrate the rapid and potent effect of the complex on the ER.

3. Minor comments in respect to text changes and data presentation:

2. Figure 1: Could the authors comment on the pronounced foci formation of the complex in kidney carcinoma (A498) cells? Alpha1 and oleate seem to perfectly co-localize at these multiple punctate structures that only seem to exist in this specific cell line shown in Figure 1H (not in the other cell lines shown), but nothing in this respect is mentioned in the text.

In response to the comment, we have analyzed the images captured by Airyscan for the different cell types that were exposed to alpha1-oleate. Consistent with the pattern in A498 cells, we observed the formation of puncta in the cytoplasm and perinuclear area in most of the cell types. We have therefore chosen to replace the panels in the Fig. 1 (C and H) with cells where such punctate staining was observed (60 min exposure). Panels A and B show fixed A549 cells exposed to alpha1-oleate for 10 minutes, where less vesiculation is seen compared to live-cell imaging captures the vesiculation better (Fig. 3). This has been clarified.

No complete (or rather very little) overlap between the single constituents (e.g. 5A) in the nucleus seems to exist. The authors should mention that mostly the single constituents can be detected, and that only limited overlap exists between the single constituents alpha1 and oleate in the nucleus in Figure 5A after 10 min of incubation with alpha1-oleate in the nucleus.

We thank the reviewer for this comment and have been quite concerned about this aspect of the Fig. 5, which signals the presence of one molecule at the time rather than a complex. As mentioned in the submitted paper (Fig. 5 legend), the process of 3D reconstruction only takes the highest signal into account. A more diffuse staining pattern or a somewhat lower signal from an equally present second molecule is not illustrated.

The figure has been replaced by confocal images that better show the topology of nuclear entry and the presence of both molecules inside the nuclei. The presence of both molecules has been quantified by line-scans (Figs. S4 and S5). Data from several additional cells has been added in the Fig. S25. The Figure has also been revised to include the Airyscan confocal images.

The title "Remodeling and nuclear entry of ER by peptide-lipid complex packages dying cell contents in nucleus" is quite hard to understand, maybe the authors could consider rephrasing?

We agree and have changed the title.

Page 3: Seem that either domain or peptide is too much in the following sentence:
"The N-terminal alpha-helical domain peptide of alpha-lactalbumin"

Has been changed.

Appendix Fig S2: The panels A-C correspond to different labeling times, but what are the three different micrographs shown for A, for B and for C? The legend does not explain.

The figures show different uptake times. Has been clarified.

There is one yellow-highlighted reference in the legend for Appendix Fig S4 left.

Has been corrected.

Information in Appendix Fig. S5 and in corresponding legend is missing in respect to the cell line shown here.

A549 cells, we apologize

The authors do not mention or discuss Fig. 2F in the text when Figure 2 is described, but describe their finding from Fig. 2F at the end of the text for Figure 3. This is confusing, please adapt either text or figure panel sequence. 2F seems to rather belong to Figure 3 (shows in principle the same as Figure 3A and B).

The Figure has been revised to include cells exposed to alpha1 peptide or oleic acid.

The organization of Figure 3 seems a bit confusing: why is the single channel for ER-tracker shown as panel 3A, the corresponding overlay as 3B and the corresponding alpha1-oleate channel in magenta in between without panel descriptor? Should this not all be 3A?

The organization of the Figure has been changed for clarity.

Figure 5: Why is the 3D model of the same cell shown in Fig. 5A and Appendix Fig 16 B (model for overlay as well as single channels combined with nucleus). Similarly, Appendix Fig. 18B shows the same 3D model of a cell as shown in 5H.

The Figure has been replaced by confocal images.

Figure 4K and L: mostly not readable. What does the colour code mean (grey, different shades of yellow, magenta, pink, blue? Similarly, gene names in 4N not readable, partially due to names overlapping. No use for gene names if they are too small to read.

We apologize, the color code is now explained in the figure legend. Regulated genes are shown as a table in Supplementary Figure 19.

Reviewer #3 (Comments to the Authors (Required)):

In this manuscript, authors show that the alpha1-oleate treatment significantly changes the shape of the nucleus, the ER, and microtubule network. Their data for this part are convincing. However, it is not convincing at all whether the ER, ribosomes, and Golgi enter the nuclei of alpha1-oleate treated cells. While this study has a potential to provide mechanistic insights into how alpha1-oleate enters nucleus and potentially changes the transcriptome, solid evidence is lacking. In my opinion this study requires substantial revisions to be published in LSA. I hope that the authors find the detailed comments below useful in revising their manuscript.

- specific major concerns essential to be addressed to support the conclusions
(i) Lack of solid evidence for the nuclear entry of the alpha1-oleate, ER, ribosomes, and Golgi.

The results suggest that there are at least two mechanisms leading to nuclear entry of the complex. One is detected as diffuse staining of the peptide and oleic acid, symmetrical and apparently unrelated to NER formation. The second phase of entry is marked by an increase in NER formation. The complex enters the nucleus enwrapped within the NER, but it remains unclear whether it ultimately reaches the nuclear lumen or remains enclosed within the ER as nuclear changes progress. Nonetheless, the NERs and the complex inside the NERs are clearly shown to be located inside the nuclear invaginations.

In Figure 5, while the authors show that alpha1-oleate signal is "inside" the nucleus in the 3D surface reconstruction, it looks like an artifact due to light scattering. In other words, because there are bright fluorescent signal at the nuclear periphery (outside of nucleus), some of the signal seems to be inside the nucleus even though it actually is not. The same applies to Figures 6 and 8, in which ER proteins, ribosome subunits, and Golgi markers appear to be enriched at the nucleoplasmic reticulum (NER). Even though it is very interesting that the ER, ribosome, and Golgi

proteins accumulate at the NER, judging from the images they seem to be outside the nucleus and the signal inside the nucleus is just scattered light from the bright perinuclear signals. If authors want to claim that they are inside the nucleus, they need to provide much more convincing data, e.g., by performing electron microscopy and show if the ER, ribosomes, and Golgi are indeed inside the nucleus.

While effects of light scattering cannot be avoided by any imaging technology except EM, we have used super-resolution Airyscan microscopy and high-end confocal microscopy in these studies. Light scattering arises primarily when light from strongly fluorescing peripheral regions is misallocated to adjacent focal planes, leading to erroneous signal localization. Confocal microscopy inherently suppresses out-of-focus light by using a pinhole aperture, ensuring that only light from the focal plane contributes to the final image. This optical sectioning capability minimizes the contribution of scattered light from bright peripheral regions to nuclear planes. The super-resolution Airyscan imaging technique uses a detector array that captures light with a high signal-to-noise ratio. Unlike traditional confocal systems, which use a single pinhole to reject out-of-focus light, Airyscan collects data from multiple points of the Airy disk and uses computational processing to reassign and enhance the resolution of the signal. This approach provides better optical sectioning, reduces out-of-focus light, and further eliminates scattering artifacts. Therefore, the volume image of the whole cell using z-stacks allows precise visualization of slices through the cell, without light scattering artifacts in both super-resolution Airyscan microscopy and diffraction-limited confocal microscopy.

The reviewer's comment prompted us to explain the data and conclusions from the images in Fig. 5 in greater detail, including at least two mechanisms leading to nuclear entry of the complex. One is detected as diffuse staining of the peptide and oleic acid, symmetrical and apparently unrelated to NER formation. With increasing NER formation, the complex enters the nuclei wrapped in NER and it's not clear if the complex finally reached the nuclear lumen or remains ER enclosed as the nuclear changes progress. None the less, the NERs and complex are clearly shown to be located inside the nuclei.

The data on organelle entry into the NER will be presented in a future paper.

(ii) Lack of control experiments to show whether only the alpha1-oleate complex can induce such morphology change or the alpha1 peptide or oleic acid alone can do.

Even though the authors show that the alpha1 peptide or oleic acid alone do not enter the cell as much as the alpha1-oleate complex does in Figure 1, such control experiments are not performed for the following experiments. Because it is possible that the peptide or oleic acid alone can trigger some signaling pathways that cause morphology changes of the nucleus, the ER, and microtubule network, the authors should perform those experiments using the peptide or oleic acid alone.

Control experiments have been included for all the results in the revised manuscript.

- minor concerns that should be addressed

Fig. 1: The authors claimed that "the alpha1 peptide and oleic acid were both detected in the cytoplasm and nuclei". It is hard to judge this from the projection images that they show. They should show single confocal slice images.

The requested information was presented in Supplementary Fig. 2.

In addition, they need to quantify the alpha1-oleate signal in the cytoplasm and nucleoplasm separately in 2D slice images instead of measuring signal in the whole cells in the projected image.

Yes, the individual channels have been added for the merged images.

Fig. 2F: It is hard to see the alpha1-oleate signal from this merged image and judge the colocalization with the ER marker. They should show images of individual channels.

The images have been replaced by panels showing the requested controls with alpha1 peptide or oleate treated cells.

Appendix Fig. 13: Atlastin and Reticulon localization do not agree with their

known/reported localization at the ER. The authors should find alternative ways to visualize those proteins, or simple remove those data.

The experiments were removed.

March 7, 2025

RE: Life Science Alliance Manuscript #LSA-2024-03114-TR

Prof. Catharina Svanborg
Lund University
Laboratory Medicine, MIG
Klinikgatan 28
BMC B13
Lund, Skane 22242
Sweden

Dear Dr. Svanborg,

Thank you for submitting your revised manuscript entitled "Rapid ER remodeling induced by a peptide-lipid complex in dying tumor cells". We would be happy to publish your paper in Life Science Alliance pending final revisions necessary to meet our formatting guidelines.

- please be sure that the authorship listing and order is correct
- please add the X and Bluesky handles of your host institute/organization as well as your own or/and one of the authors in our system
- we encourage you to revise the figure legends for figures S1, S7, and S8 such that the figure panels are introduced in alphabetical order
- please be sure to add call-outs for all panels in the supporting figures to the manuscript text

FIGURE CHECK:

- please add sizes next to the blots in figure 4F, S21E and F
- There appears to be a splice after the first lanes in the blots for p-eIF2a and GAPDH in figure S21E. Please indicate the splice with a vertical black line, and mention what the line signifies in the figure legend. Please also provide these 2 blots as Source Data for our review.

A. FINAL FILES:

B. MANUSCRIPT ORGANIZATION AND FORMATTING:

Sincerely,

March 13, 2025

RE: Life Science Alliance Manuscript #LSA-2024-03114-TRR

Prof. Catharina Svanborg
Lund University
Laboratory Medicine, MIG
Klinikgatan 28
BMC B13
Lund, Skane 22242
Sweden

Dear Dr. Svanborg,

Thank you for submitting your Research Article entitled "Rapid ER remodeling induced by a peptide-lipid complex in dying tumor cells". It is a pleasure to let you know that your manuscript is now accepted for publication in Life Science Alliance. Congratulations on this interesting work.

DISTRIBUTION OF MATERIALS:

Again, congratulations on a very nice paper. I hope you found the review process to be constructive and are pleased with how the manuscript was handled editorially. We look forward to future exciting submissions from your lab.

Sincerely,
